# Balancing safety and efficiency in human decision-making

**Pranav Mahajan[1]\*, Shuangyi Tong[2], Sang Wan Lee[3,4,5], Ben Seymour[1,2]\***

[1]Wellcome Centre for Integrative Neuroimaging, FMRIB, Nuffield Department of Clinical Neurosciences, University of Oxford, Oxford, United Kingdom; [2]Institute of Biomedical Engineering, University of Oxford, Oxford, United Kingdom; [3]Department of Brain and Cognitive Sciences, Korea Advanced Institute of Science and Technology (KAIST), Daejeon, Republic of Korea; [4]Kim Jaechul Graduate School of AI, KAIST, Daejeon, Republic of Korea; [5]KAIST Center for Neuroscience-inspired Artificial Intelligence, Daejeon, Republic of Korea

**\*For correspondence:**
pranav.mahajan@ndcn.ox.ac.uk (PM);
ben.seymour@ndcn.ox.ac.uk (BS)

**Competing interest:** The authors declare that no competing interests exist.

## eLife Assessment

This **important** work describes results from a set of simulation and empirical studies of a set-up assessing exploratory behavior in a potentially rewarding environment that contains danger. The core idea is that an instrumental agent can be helped to be both effective and safe, thus avoiding excessive danger, during exploratory behavior, if the influence of an independent Pavlovian fear is flexibly gated based on uncertainty. This work is grounded in previous foundational work on Pavlovian control of instrumental choice, and significantly extends prior work showing that the impact of Pavlovian reward biases can be flexibly gated. The conclusion that safe but effective exploration can be achieved based on a flexibly weighted combination of a Pavlovian and an instrumental agent is **convincing**.

**Abstract** The safety-efficiency dilemma describes the problem of maintaining safety during efficient exploration and is a special case of the exploration-exploitation dilemma in the face of potential dangers. Conventional exploration-exploitation solutions collapse punishment and reward into a single feedback signal, whereby early losses can be overcome by later gains. However, the brain has a separate system for Pavlovian fear learning, suggesting a possible computational advantage to maintaining a specific fear memory during exploratory decision-making. In a series of simulations, we show this promotes safe but efficient learning and is optimised by arbitrating Pavlovian avoidance of instrumental decision-making according to uncertainty. We provide a basic test of this model in a simple human approach-withdrawal experiment in virtual reality and show that this flexible avoidance model captures choice and reaction times. These results show that the Pavlovian fear system has a more sophisticated role in decision-making than previously thought, by shaping flexible exploratory behaviour in a computationally precise manner.

## Introduction

Humans and animals inhabit a complex and dynamic world where they need to find essential rewards such as food, water, and shelter, whilst avoiding a multitude of threats and dangers which can cause injury, disability, or even death. This illustrates a tension at the heart of learning and decision-making systems: on the one hand, one wants to minimise environmental interactions required to learn to acquire rewards (be sample efficient), but on the other hand, it is important not to accrue excessive damage in the process - which is particularly important if you only get one chance at life. This

**eLife digest** Animals need to navigate a complex world where they gain rewards, such as finding food and shelter, while avoiding danger on a daily basis. This creates a 'safety-efficiency dilemma' in how to pursue rewards efficiently without increasing the risk of harm.

The human brain relies on two key systems to manage this challenge. The instrumental system learns from the consequences of our actions, weighing costs and benefits to maximize overall outcomes. In contrast, the Pavlovian fear system generates automatic defensive responses. For example, it learns to associate certain cues with danger and triggers instinctual reactions, like pulling the hand away from a hot stove. Sometimes these systems conflict—fear urges withdrawal even when careful action might lead to a greater reward—and the brain must resolve this tension. Mahajan et al. set out to explore how the brain flexibly manages the influence of this automatic fear system depending on how uncertain a situation is.

Is it computationally advantageous for the brain to adjust the impact of Pavlovian fear based on outcome uncertainty? And is there evidence that humans actually use this strategy? These questions are important because the influence of fear has often been treated as fixed. Recognizing its flexibility offers a more nuanced understanding of how we balance safety and efficiency in decision-making.

Mahajan et al. found that a flexible Pavlovian fear system, guided by uncertainty, promotes safer decisions without substantially sacrificing efficiency. The researchers first used computer simulations to show that a model in which fear's influence decreases as the environment becomes more predictable resolves the safety–efficiency trade-off more effectively than models with a fixed fear response.

They then tested this prediction in a virtual reality experiment, where human participants decided whether to approach or withdraw from stimuli that could lead to a mild electric shock. The results showed that the flexible model best explained both people's choices and their reaction times.

These findings may eventually benefit individuals with anxiety disorders or chronic pain, conditions often marked by excessive avoidance behaviors. The results suggest the core issue may not simply be an overactive fear system, but rather a difficulty in flexibly reducing its influence when situations become predictable. This perspective could inspire therapies that aim to help patients to better distinguish between controllable and uncontrollable threats, making their defensive responses more adaptable – a concept already present in some forms of cognitive behavioral therapy.

safety-efficiency dilemma is related to the exploration-exploitation dilemma, in which the long-term benefits of information acquisition are balanced against the short-term costs of avoiding otherwise valuable options. Most solutions to the exploration-exploitation dilemma consider things only from the point of view of a single currency of reward, and hence, early losses can be overcome by later gains. Thus, many engineering solutions involve transitioning from exploratory strategies to more exploitative strategies over time as an agent gets more familiar with the environment. However, such solutions could be insufficient if some outcomes are incommensurable with others; for instance, damage accrues to the point that cannot be overcome, or worse still, leads to system failure 'death' before you ever get the chance to benefit through exploitation, emphasising the need for safe (early) exploration. Safe learning (*García and Fernández, 2015*) is an emerging topic in artificial intelligence and robotics, with the advent of adaptive autonomous control systems that learn primarily from experience: e.g., robots intended to explore the world without damaging or destroying themselves (or others) - the same concern animals and humans have.

A biological solution to this problem may be to have distinct systems for learning, for instance, having Pavlovian reward and punishment systems in addition to an instrumental system, which can then be integrated together to make a decision (*Bach and Dayan, 2017*; *Elfwing and Seymour, 2017*). A dissociable punishment system could then allow, for example, setting a lower bound on losses which must not be crossed during early learning. The brain seems likely to adopt a strategy like this since we know that Pavlovian fear processes influence instrumental reward acquisition processes (e.g. in paradigms such as conditioned suppression [*Kamin et al., 1963*] and Pavlovian-instrumental transfer [*Prévost et al., 2012*; *Talmi et al., 2008*]). However, it is not clear if this exists as a static system, with a constant Pavlovian influence over instrumental decisions, or a flexible system in which the Pavlovian influence is gated by information or experience. Computationally, it implies a

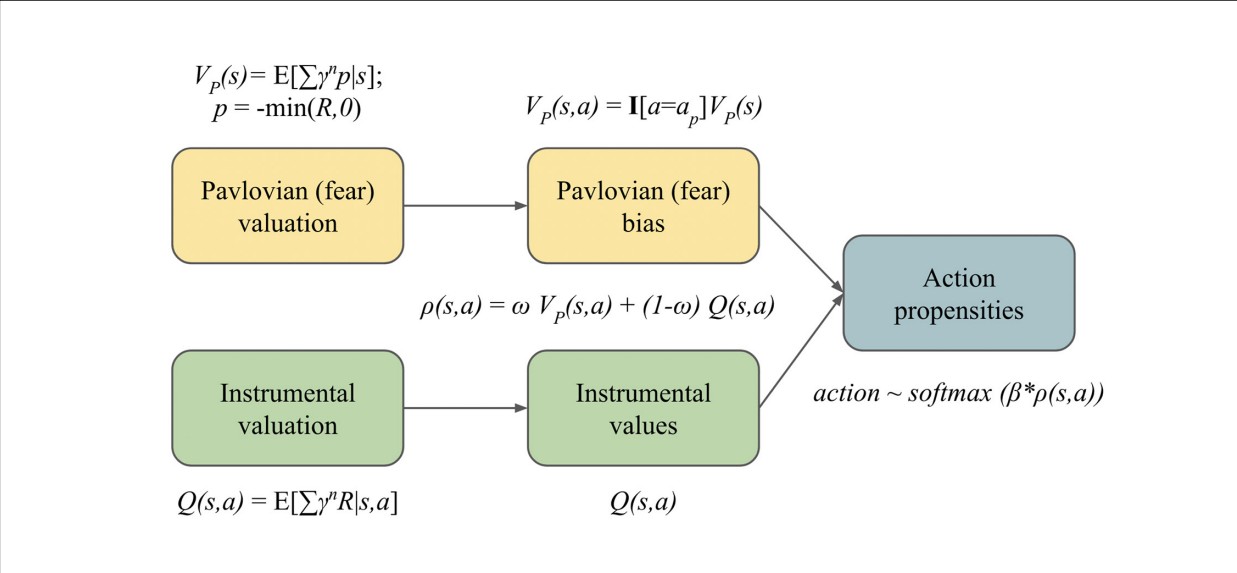

**Figure 1.** An illustration of Pavlovian Avoidance Learning (PAL) model. Pavlovian and instrumental valuations are combined to arrive at action propensities used for (softmax) action selection. The Pavlovian bias influences protective behaviours through safer (Boltzmann) exploration, and the arbitration between the Pavlovian and instrumental systems is performed using the parameter $\omega$. Here, $R$ denotes the feedback signal which can take both positive values (in the case of rewards) and negative values (in the case of punishments). Please see Materials and methods for technical details; notations for the illustration follow *Dorfman and Gershman, 2019*.

multi-attribute architecture involving modular systems that separately learn different components of feedback (rewards, punishments) with the responses or actions to each then combined.

In this paper, we ask two central questions: (1) whether it is computationally (normatively) adaptive to have a flexible system that titrates the influence of 'fear' based on uncertainty, i.e., reduces the impact of fear after exploration and (2) whether there is any evidence that humans use this sort of flexible meta-control strategy. We first describe a computational model of how Pavlovian (state-based) responses shape instrumental (action-based control) processes and show how this translates to a multi-attribute reinforcement learning (RL) framework at an algorithmic level. We propose how Pavlovian-instrumental transfer may be flexibly guided by an estimate of outcome uncertainty (*Bach and Dolan, 2012*; *Dorfman and Gershman, 2019*) - which effectively acts as a measure of uncontrollability. We use Pearce-Hall associability (*Krugel et al., 2009*), which is an implementationally simple and direct measure of uncertainty that has been shown to correlate well with both fear behaviour (skin conductance) and brain (amygdala) activity in fear learning studies (*Li et al., 2011*; *Zhang et al., 2016*; *Zhang et al., 2018*). Below, we demonstrate the safety-efficiency trade-off in a range of simulation environments and show how it can be solved with a flexible Pavlovian fear bias. Consequently, we then test basic experimental predictions of the model in a virtual reality-based approach-withdrawal task involving pain, which builds upon previous Go-No Go studies studying Pavlovian-instrumental transfer (*Cavanagh et al., 2013*; *Guitart-Masip et al., 2012*). The virtual-reality approach confers a greater ecological validity, and the immersive nature may contribute better to fear conditioning, making it easier to distinguish the aversive components.

## Results

### Pavlovian avoidance learning model

Our model consists of a Pavlovian punishment (fear) learning system and an integrated instrumental learning system (*Figure 1*). The standard (rational) RL system is modelled as the instrumental learning system. The additional Pavlovian fear system biases the withdrawal actions to aid in safe exploration, in line with our hypothesis. The Pavlovian system learns punishment expectations for each stimulus/ state, with the corresponding Pavlovian responses manifest as action propensities to withdraw. For simplicity, we don't include a Pavlovian reward system, or other types of Pavlovian fear response

(*Bolles, 1970*). The instrumental system learns combined reward and punishment expectations for each stimulus-action or state-action pair and also converts these into action propensities. Both systems learn using a basic temporal difference updating rule (or in instances, its special case, the Rescorla-Wagner [RW] rule). The ultimate decision that the system reaches is based on integrating these two action propensities, according to a linear weight, $\omega$. Below, we consider fixed and flexible implementations of this parameter, and test whether a flexible $\omega$ confers an advantage. We implement the flexible $\omega$ using Pearce-Hall associability (see *Equation 15* in Materials and methods). The Pearce-Hall associability maintains a running average of absolute temporal difference errors ($\delta$) as per *Equation 14*. This acts as a crude but easy-to-compute metric for outcome uncertainty which gates the influence of the Pavlovian fear system, in line with our hypothesis. This implies that the higher the outcome uncertainty, as is the case in early exploration, the more cautious our agent will be, resulting in safer exploration. For simulations, we use standard grid world-like environments, which provide a didactic tool for understanding Pavlovian-instrumental interactions (*Dayan et al., 2006*). Since Pavlovian biases influence not only choices but also reaction times (RTs), we extend our model to reinforcement learning diffusion decision-making (RLDDM) models (*Fengler et al., 2022*; *Fontanesi et al., 2019*; *Pedersen et al., 2017*).

## Experiment 1: A simulated flexible fear-commissioning model balances safety and efficiency

We consider a simple fully-observable grid world environment with stochastic state transitions and fixed starting state and fixed rewarding goal state. *Figure 2A* illustrates how the misalignment of Pavlovian bias and instrumental action can lead to a safety-efficiency dilemma. The Pavlovian action is assumed to be an evolutionarily acquired simple withdrawal response (*Figure 2B*), and the Pavlovian state value is learned during the episode and shapes instrumental policy and value acquisition (*Figure 2C*). *Figure 2C* shows value plots for the instrumental policy with and without a Pavlovian bias. All plots show values and policy at the end of 1000 episodes of learning. These heatmaps denote value, i.e., the expectation of cumulative long-term rewards $R$ (including any punishments) and the arrows show the policy, i.e., actions that maximise this value. Additionally, the learned punishment value $V_p$ of the Pavlovian bias is also shown along with the Pavlovian avoidance learning (PAL) policy. The PAL value function and policy shown in *Figure 2C* utilises the flexible $\omega$ scheme utilised below.

*Figure 2D* plots cumulative pain accrued over multiple episodes during learning and is our measure of safety. *Figure 2E* plots cumulative steps or environment interactions over episodes and is our measure of sample efficiency. Here, sampling efficiency is represented by the total number of environment interactions or samples required to reach the rewarding goal which terminates the episode. Simply, if an agent requires more samples to reinforce and acquire the rewarding goal, it is less efficient.

The simulation results with a fixed Pavlovian influence (*Figure 2D and E*) show that adding a Pavlovian fear system to the instrumental system makes it safer in the sense that it achieves the goal of solving the environment while accruing lesser cumulative pain over episodes. However, we observe that as the influence of the Pavlovian fear system increases, with an increase in $\omega$, it achieves safety at the expense of sample efficiency (within reasonable bounds such as until $\omega$=0.5). Whereas under very high Pavlovian fear influence ($\omega$=0.9), the agent loses sight of the rewarding goal and performs poorly in terms of both safety and efficiency as the episode doesn't terminate until it finds the goal.

However, the flexible omega policy (with $\alpha_\Omega = 0.6$ and $\kappa = 6.5$) achieves safety almost comparable to $\omega$=0.5, which is the safest fixed $\omega$ policy amongst $\omega$=0.1, 0.5, 0.9 at a much higher efficiency than $\omega$=0.5, 0.9, thus improving the safety-efficiency trade-offs (*Figure 2F*). In this way, the PAL model encourages cautious exploration early on when uncertainty is higher and reduces the Pavlovian biases as the uncertainty is resolved (*Figure 2F*). The flexible $\omega$ value at convergence depends on the environment statistics: transition probabilities and reward/punishment magnitudes. We utilise a simple linear scaling of associability clipped at 1 to arrive at arbitrator $\omega$ (*Equation 15*) instead of another alternative such as sigmoid to avoid additional unnecessary meta parameters (i.e. bias shift) to be tuned. In this environment, the value at convergence is $\omega$=0.42, due to some irreducible uncertainty in state transitions (10% chance of incorrect transition). The differences in learned instrumental value functions between PAL and a purely instrumental agent are visible in *Figure 2C* showing how the Pavlovian bias sculpts instrumental value acquisition.

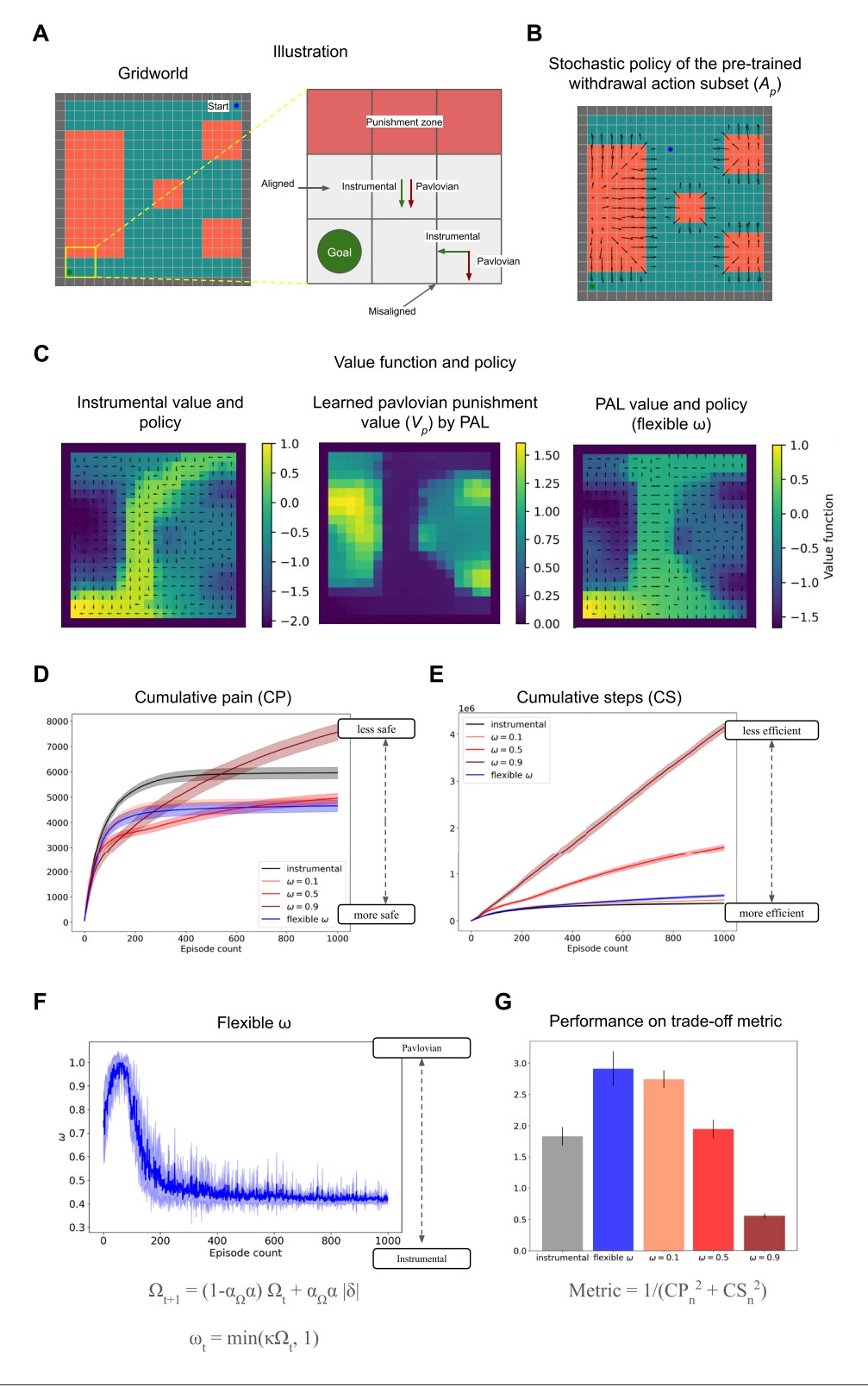

**Figure 2.** Demonstration of safety-efficiency trade-off and the flexible arbitration scheme in a grid world environment. (**A**) Grid world environment with starting state in the top-right corner and rewarding goal state ($R = +1$) in the bottom-left corner and the red states are painful ($R = -0.1$). The grid world layout follows *Gehring and Precup, 2013*. Inset provides a didactic example of misalignment between Pavlovian bias and instrumental

*Figure 2 continued on next page*

*Figure 2 continued*

action. (**B**) Stochastic policy of pre-trained withdrawal action subset $A_p$, which is biased with Pavlovian punishment values in the Pavlovian avoidance learning (PAL) agent. (**C**) The learned instrumental values and Pavlovian fear bias $V_p$(heatmap) and policy (arrows) are learned by the instrumental and flexible $\omega$ agent by the end of the learning duration. The value functions plotted are computed in an on-policy manner. (**D**) Cumulative pain accrued by fixed and flexible $\omega$ agents whilst learning over 1000 episodes as a measure of safety averaged over 10 runs. (**E**) Cumulative steps required to reach the fixed goal by fixed and flexible $\omega$ agents whilst learning over 1000 episodes as a measure of sample efficiency, averaged over 10 runs (**F**) Plot of flexibly modulated $\omega$ arbitration parameter over the learning duration averaged over 10 runs. This shows a transition from a higher Pavlovian bias to a more instrumental agent over episodes as learning about the environment reduces uncertainty. (**G**) Comparison of different agents using a trade-off metric and to be used only for didactic purposes (using *Equation 16* and more details in Materials and methods).

In the Appendix, we provide additional simulations that show the robustness of these results with respect to metaparameters $\alpha_\Omega$ and $\kappa$ (*Appendix 1—figure 1*), environments in which the reward locations vary (*Appendix 1—figure 2*), and other grid world environments (*Appendix 1—figure 3*).

## Experiment 2: Constant Pavlovian bias introduces sampling asymmetry and affects instrumental value propagation

Observing the differences in the on-policy value functions with and without the Pavlovian influence (*Figure 2*) prompted us to further tease apart the effect of constant Pavlovian bias on sampling asymmetry, and consequent differences in instrumental value discovery and value propagation through the states. We investigated how different fixed values of $\omega$ can lead to sampling asymmetry, which refers to exploration where certain states are visited or sampled unevenly compared to others. In this set of results, we wish to qualitatively tease apart the role of a Pavlovian bias in shaping and sculpting the instrumental value and also provide more insight into the resulting safety-efficiency trade-off. Having shown the benefits of a flexible $\omega$ in the previous section, here we only vary the fixed $\omega$ to illustrate the effect of a constant bias and are not concerned with the flexible bias in this experiment.

We tested agents with different fixed $\omega$ in two simulated environments: (1) A T-maze and (2) a three-route task. The T-maze task environment (*Figure 3A*) has asymmetric rewards ($R = +0.1$ on the left, whereas $R = +1$ on the right). However, the agent will have to walk through a patch of painful states to reach the larger goal on the right; even the safest path will incur a damage of at least $R = -0.5$ or worse. Taking discounting into account, the goal on the right is marginally better than the one on the left, and the instrumental agent achieves both of the goals nearly an equal number of times (*Figure 3B*). Comparing the instrumental agent with other agents in *Figure 3C* shows diminished positive (reward) value propagation leading to the $R = 1$ goal on the right as the constant Pavlovian bias increases, showing how such sampling asymmetry can prevent value discovery of states leading to $R = 1$ goal. The safety efficiency trade-off can also be observed through *Figure 3B*. This illustrates one of the main tenets of our model - that having a Pavlovian fear system ensures a separate 'un-erasable' fear/punishment memory which makes the agent more avoidant to punishments. This is helpful in softly ensuring an upper bound on losses, by (conservatively) foregoing decisions resulting in immediate loss, but followed by much larger rewards. This is where the safety-efficiency trade-off marks a clear distinction from the exploration-exploitation trade-off, in which earlier losses can be overcome by gains later on.

The three-route task simulation includes three routes with varying degrees of punishments (*Figure 3D*), inspired by previous manipulandum tasks (*Glogan et al., 2021*; *Meulders et al., 2016*; *van Vliet et al., 2020*; *van Vliet et al., 2021*). We observe that increasing the constant Pavlovian bias up until $\omega$=0.7 leads to increased safety (*Figure 3E*). Beyond $\omega$=0.7, a high fixed Pavlovian bias may incur unnecessarily high cumulative pain and steps as its reward value propagation is diminished (*Figure 3G*) and attempts to restrict itself to pain-free states (*Figure 3F*) whilst searching for reward (despite stochastic transitions which may lead to slightly more painful encounters in the long run). Comparing the cumulative state visit plots of *Figure 3F*, the instrumental agent with an agent with high constant Pavlovian bias $\omega$=0.9, we observe that the latter showed an increased sampling of the states on the longest route with no punishments. Comparing the value function plots (*Figure 3G*), we observe that a high constant Pavlovian bias impairs the value propagation (it is more diffused) of the

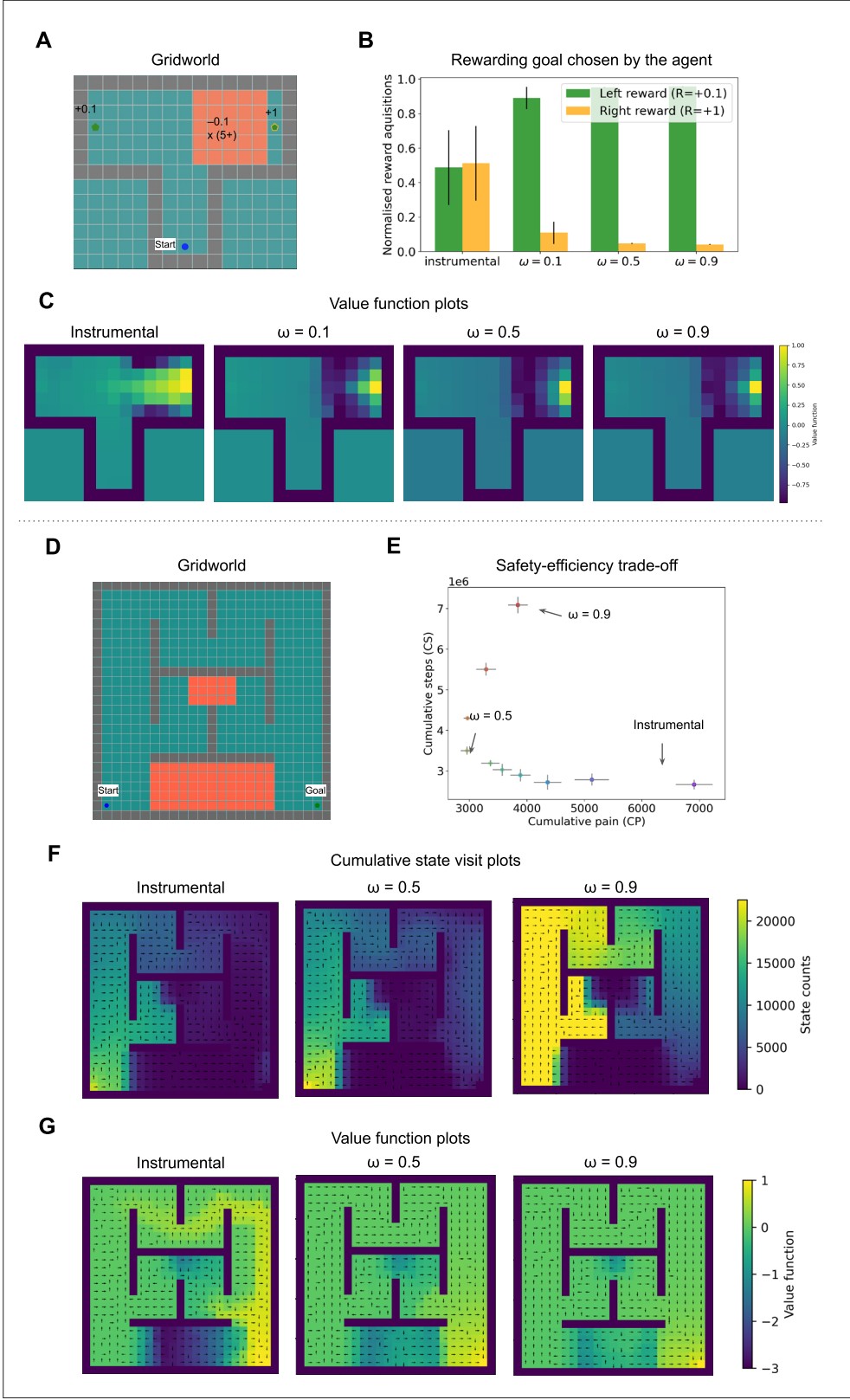

**Figure 3.** Demonstration of sampling asymmetry due to constant Pavlovian bias. (**A**) T-maze grid world environment with annotated rewards and punishments. (**B**) Proportion of the rewarding goal chosen by the agent. (**C**) Value function plots for $\omega$=0,0.1,0.5,0.9 show diminished value propagation from the reward on the right. (**D**) Grid world environment with three routes with varying pain. (**E**) Cumulative steps required to reach the goal vs

*Figure 3 continued on next page*

*Figure 3 continued*

cumulative pain accrued by fixed $\omega$ agents ranging from $\omega$=10 to $\omega$=0.9. (**F**) State visit count plots for $\omega$=0, 0.5, 0.9, i.e., instrumental and constant Pavlovian bias agents. (**F**) Value function plots for $\omega$=0, 0.5, 0.9.

rewarding goal in comparison to an instrumental agent. Such high levels of constant Pavlovian bias can be a model of maladaptive anxious behaviour.

In conclusion, the simulations with this environment show that the Pavlovian fear system can assist in avoidance acquisition; however, a constant Pavlovian bias, depending on the degree of bias, leads to sampling asymmetry and impaired value propagation.

*Appendix 1—figure 3* includes the performance comparison of agents with a suitable flexible $\omega$ and with fixed $\omega$ values on the three-route task. *Appendix 1—figure 4* shows the results of a human experiment with subjects navigating a three-route virtual reality maze similar to the simulated one.

## Experiment 3: Human approach-withdrawal conditioning is modulated by outcome uncertainty

Our first experiment showed the benefit of having an outcome uncertainty-based flexible $\omega$ arbitration scheme in balancing safety and efficiency, in a series of grid worlds. In this next experiment, we aimed to find basic evidence that humans employ such a flexible fear commissioning scheme. This is not intended as an exhaustive test of all predictions of the model, but to show in principle that there are situations in which a flexible, rather than fixed Pavlovian influence, provides a good fit to real behavioural data. In line with our grid world simulations, we expected a Pavlovian bias in choices, but in addition to it, we also expected a Pavlovian bias in RTs.

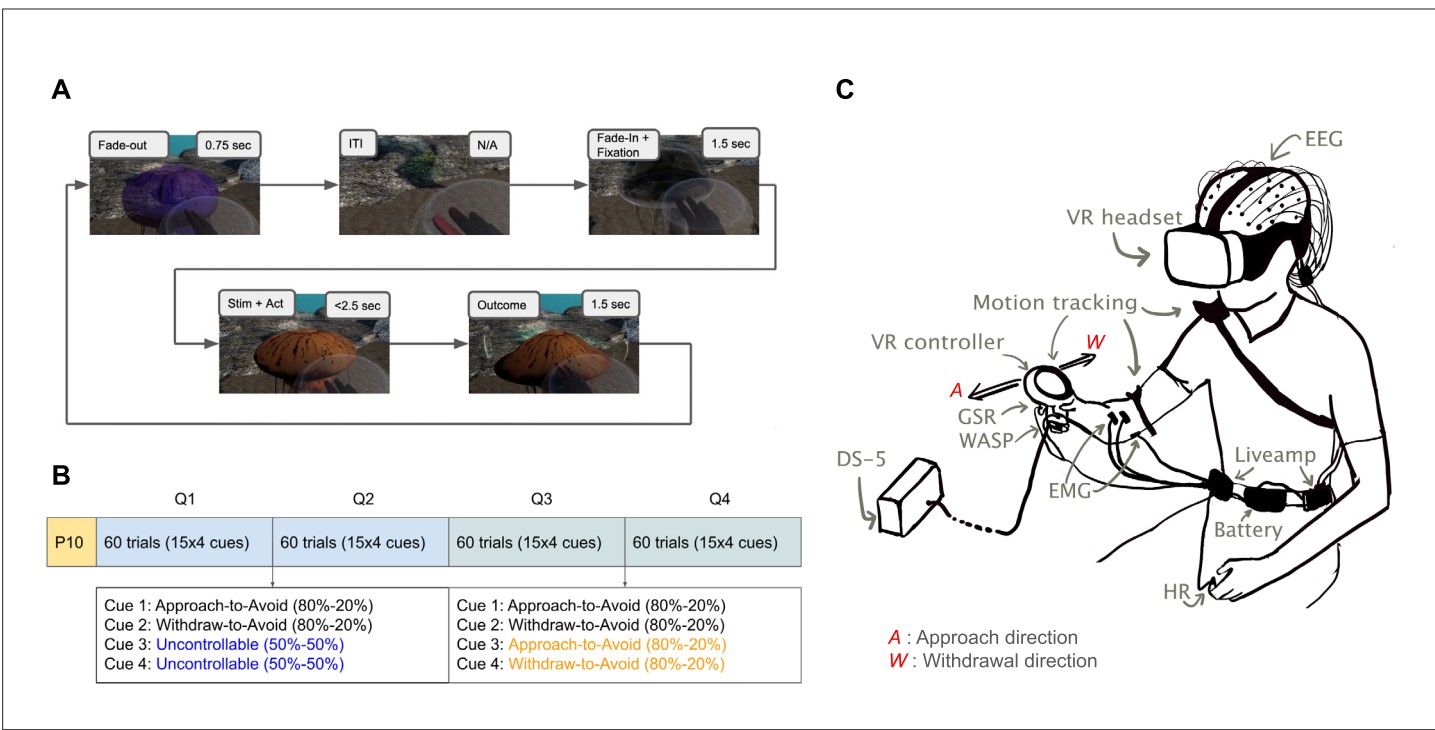

**Figure 4.** An illustration of the VR Approach-Withdrawal task and trial and block protocols. (**A**) Trial protocol: The participant is expected to take either an approach action (touch the jellyfish) or withdrawal action (withdraw the hand towards oneself) within the next 2.5 s once the jellyfish changes colour. The participant was requested to bring their hand at the centre of a bubble located halfway between the participant and the jellyfish to initiate the next trial where a new jellyfish would emerge (video). (**B**) Block protocol: First half of the trials had two uncontrollable cues and two controllable cues, and the second half had all controllable cues with aforementioned contingencies. The main experiment 240 trials were preceded by 10 practice trials which do not count towards the results. (**C**) Illustration of experimental setup. VR: virtual reality, WASP: surface electrode for electrodermal stimulation, DS-5: constant current stimulator, GSR: galvanic skin response sensors, HR: heart rate sensor, EMG: electromyography sensors, EEG: electroencephalogram electrodes, Liveamp: wireless amplifier for mobile EEG.

*Figure 4* describes the trial protocol (*Figure 4A*), block protocol (*Figure 4B*), and experimental setup (*Figure 4C*). We conducted a VR-based approach-avoidance task (28 healthy subjects, of which 14 females and average age 27.96 years) inspired by previous Go-No Go task studies for isolating Pavlovian bias, especially its contributions to misbehaviour (*Cavanagh et al., 2013*; *Dorfman and Gershman, 2019*; *Gershman et al., 2021*; *Guitart-Masip et al., 2012*; *Mkrtchian et al., 2017a*; *Mkrtchian et al., 2017b*). The subject's goal was to make a correct approach or withdrawal decision to avoid pain, with four different cues associated with different probabilities of neutral or painful outcomes. We expected the Pavlovian misbehaviour to cause incorrect withdrawal choices for cues where the correct response would be to approach. And in terms of RTs, we expected the bias to slow down correct approach responses and speed up correct withdrawal responses. We explicitly attempted to change the outcome uncertainty or controllability, in a similar way to previous demonstrations (*Dorfman and Gershman, 2019*), but with controllability changing *within* the task. To do this, we set up two of the four cues to be uncontrollable in the first half (i.e. outcome is painful 50% of the times regardless of the choice), but which then become controllable in the second half (i.e. the correct choice will avoid the pain 80% of the times). We anticipated that the Pavlovian bias in choice and RTs would be modulated along with the change in uncontrollability. The virtual reality environment improves ecological validity (*Parsons, 2015*) and introduces gamification, which is known to also improve reliability of studies (*Kucina et al., 2023*; *Sailer et al., 2017*; *Zorowitz et al., 2023*), which is important in attempts to uncover potentially subtle biases.

We observe that all subjects learn to solve the task well and solve it better than chance (i.e. fewer than 120 shocks in 240 trials). Out of 240 trials, they receive 88.96 shocks on average (std. deviation = 12.62). We first attempted to test our hypotheses using behavioural metrics of Pavlovian withdrawal bias in choices and RTs. However, our behavioural choice-based metrics cannot distinguish a random exploratory action from Pavlovian misbehaviour. Further, it cannot account for effects of a non-Pavlovian baseline bias $b$. Thus, we did not find any statistically significant result due to noisy behavioural metrics, results, and more information provided in *Appendix 1—figure 5*.

We next aimed to test our hypotheses by model comparison of RL models (*Figure 5*) and RLDDM models (*Figure 5E*) which guides our results below. We used a hierarchical Bayesian estimation of model parameters to increase the reliability across tasks. We found that the baseline action bias $b$, instrumental learning, and the Pavlovian withdrawal bias competed for behavioural control, as observed in previous studies (*Cavanagh et al., 2013*; *Guitart-Masip et al., 2012*) (parameter distribution plots in *Appendix 1—figures 6–9*). However, unlike previous studies that have treated Pavlovian bias as fixed, we found that the flexible Pavlovian bias better explained the behavioural data; please see *Figure 5B and F*.

Similar to *Guitart-Masip et al., 2012*, the simple RW learning model represented the base model. RW+bias includes a baseline bias $b$ that can take any positive or negative value; positive value denoting a baseline bias for approach and negative denoting a baseline bias for withdrawal. From group-level and subject-level parameter distribution plots (*Appendix 1—figures 6–9*), we observe that this baseline bias is for approach for most subjects. This is in line with previous studies (*Cavanagh et al., 2013*; *Guitart-Masip et al., 2012*) and as suggested by our data showing a significant baseline difference in the number of approaches and withdrawal actions across all subjects and trials (*Appendix 1—figure 5A*). Note that here, this baseline bias is not learned as it is with a Pavlovian bias. RW+bias+Pavlovian (fixed) model includes a fixed Pavlovian bias and is most similar to models by *Cavanagh et al., 2013*; *Guitart-Masip et al., 2012*, which also used reward and punishment sensitivities for the instrumental learning but did not scale the instrumental values by $(1 - \omega)$ as done in our model. Our models do not have reward and punishment sensitivities. From group-level and subject-level parameter distribution plots (*Appendix 1—figures 6–9*), we observe that the distribution of fixed $\omega$ is significantly positive and non-zero. RW+bias+Pavlovian (flexible) model includes a flexible Pavlovian bias as per our proposed associability-based arbitration scheme. We found that the flexible $\omega$ model fits significantly better than the fixed $\omega$ model (*Figure 5B*), i.e., the flexible $\omega$ model has the lowest leave-one-out information criteria (LOOIC) score amongst models compared. By comparing incremental improvements in LOOIC, we observe that adding the baseline bias term leads to the most improvement in model fit, followed by changing the fixed $\omega$ to a flexible $\omega$ scheme. Here, we plot LOOIC for model comparison, but *Appendix 1—tables 1 and 2* include both LOOIC and Watanabe-Aikake information criterion (WAIC) scores, showing the same result. Further, it can be seen that $\omega$ tracks associability,

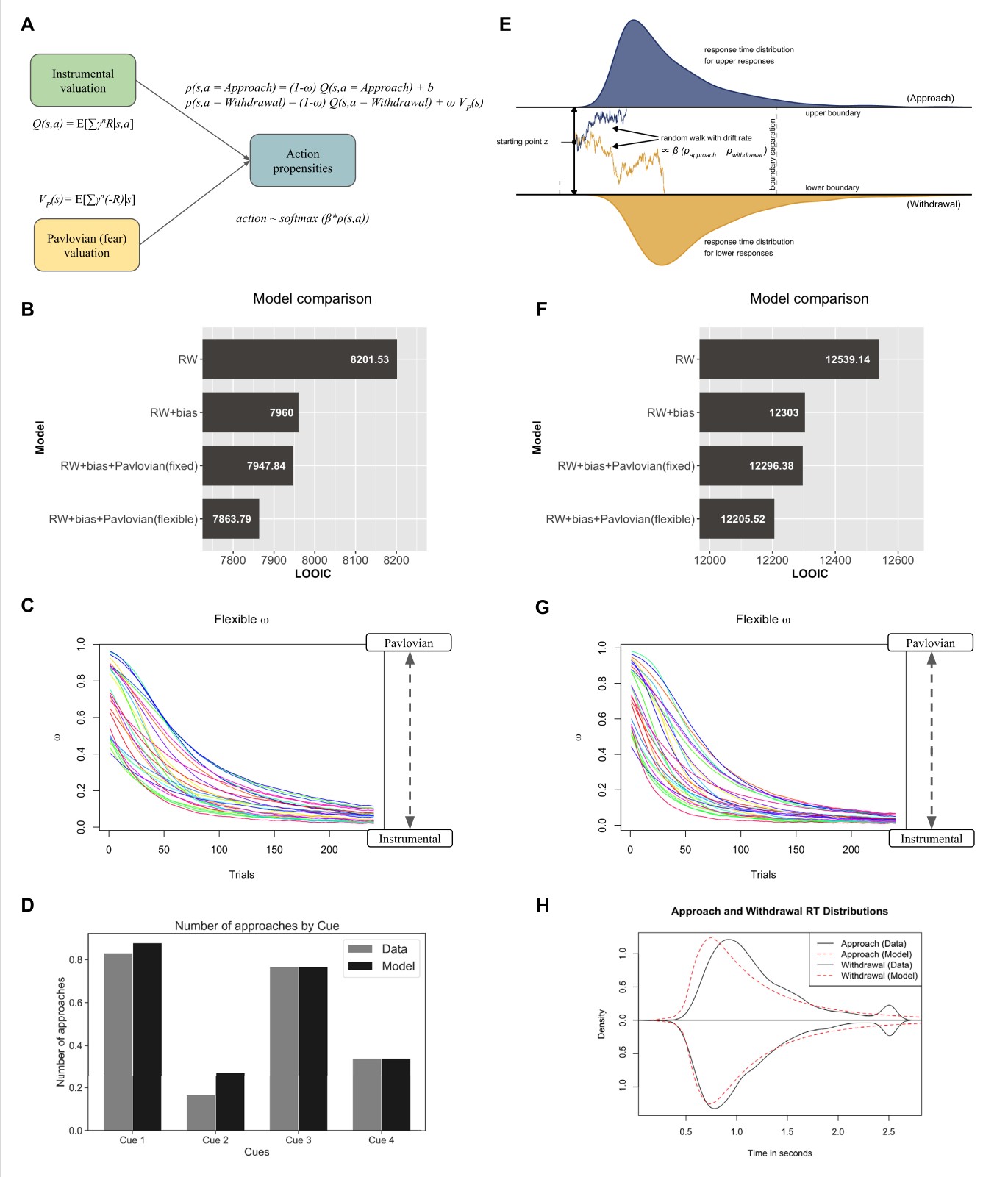

**Figure 5.** RL and RLDDM model fitting results on VR Approach-Withdrawal task. The left panels show choice model fit results using reinforcement learning (RL) models. The right panels show choice and reaction times (RTs) model fit results using reinforcement learning diffusion decision-making (RLDDM) models. (**A**) Simplified RL model from *Figure 1* for the approach-withdrawal task. (**B**) Model comparison shows that the model with flexible Pavlovian bias fits best to choices in terms of leave-one-out information criteria (LOOIC). (**C**) Flexible $\omega$ from the RL model over 240 trials for 28

*Figure 5 continued on next page*

*Figure 5 continued*

participants. (**D**) Number of approaches aggregated over all subjects and all trials in data and model predictions by the RL model with flexible.$\omega$ (normalised to 1). (**E**) Simplified illustration of RLDDM for the approach-withdrawal task, where the baseline bias $b$ and the Pavlovian bias $\omega V_p(s)$ is also included in the drift rate. (The base figure is reproduced from Figure 8 from *Desch et al., 2022* with modifications.) (**F**) Model comparison shows that the model with flexible Pavlovian bias fits best to choices and RTs in terms of LOOIC. (**G**) Flexible $\omega$ from the RLDDM over 240 trials for 28 participants. (**H**) Distribution of approach and withdrawal RTs aggregated over all subjects and trials in data and model predictions by the RLDDM with flexible $\omega$. The bump in RTs at 2.5 s is because of timeout (inactive approaches and withdrawals, please see *Appendix 1—figure 5*).

which decreases over the trials (*Figure 5C*) (which also resembles *Figure 2E*). *Figure 5D* shows a plot comparing the number of approaches (normalised to 1), aggregated over all subjects and trials by cue types for data and the best fitting model predictions. We observe qualitatively that the subjects learn to perform the correct actions for each cue and that the model predictions qualitatively reproduce the data.

We then extend the model fitting to also incorporate RTs, using an RLDDM (*Desch et al., 2022*; *Fontanesi et al., 2019*; *Pedersen et al., 2017*). The propensities calculated using the RL model are now used as drift rates in the DDM, and the RTs are calculated using a Wiener distribution for a diffusion-to-bound process. Thus, the drift rate is proportional to the difference in propensities between approach and withdrawal action. Since Pavlovian bias is also dependent on punishment value, similar to instrumental values, we included the Pavlovian bias and the baseline bias in the drift rate. Thus, the best propensity for an action in choice selection in RL models drives the drift rate in our RLDDM models. We found that the RLDDM replicates the results for model fitting (*Figure 5E*) and flexible $\omega$ (*Figure 5F*). *Figure 5H* shows the distribution for approach and withdrawal RTs aggregated over all subjects, over all trials, in data and model predictions. The data shows that the withdrawal RTs are slightly faster than approach RTs (*Figure 5H*) and that the best fitting model captures the withdrawal RT distribution well, but can be improved in the future to capture approach RT distribution better.

*Appendix 1—figure 5* includes behavioural results for the experiment data. *Appendix 1—figures 6–9* include group-level and subject-level (hierarchically fitted) model parameter distributions. *Appendix 1—table 1* and *Appendix 1—table 2* mention model parameters with LOOIC and WAIC values for all RL models and RLDDM models. We observed that all Rhat values were strictly less than 1.05 (most parameters were less than 1.01 and generally close to 1), indicating that the models had converged.

## Discussion

In summary, this paper shows that addition of a fear-learning system, implemented as a Pavlovian controller in a multi-attribute RL architecture, improves safe exploratory decision-making with little cost of sample efficiency. Employing a flexible arbitration scheme where Pavlovian responses are gated by outcome uncertainty (*Bach and Dolan, 2012*) provides a neurally plausible approach to solving the safety-efficiency dilemma. Our experimental results support the hypothesis of such a flexible fear commissioning scheme and suggest that inflexible Pavlovian bias can explain certain aspects of maladaptive 'anxious' behaviour (please see *Appendix 1—figure 10*). This can be helpful in making novel predictions in clinical conditions, including maladaptive persistent avoidance in chronic pain in which it may be difficult to 'unlearn' an injury.

Broadly, our model sits amidst the landscape of safe RL (*García and Fernández, 2015*). In principle, it can be viewed through the lens of constrained Markov decision processes (*Altman, 1999*), where the Pavlovian fear system is dedicated towards keeping constraint violations at a minimum. In the realm of safe learning, there exists a dichotomy: one can either apply computer science-driven approaches to model human and animal behaviour, as seen in optimising worst-case scenarios (*Heger, 1994*) and employing beta-pessimistic $Q$-learning (*Gaskett, 2003*) for modelling anxious behaviour (*Zorowitz et al., 2020*), or opt for neuro-inspired algorithms and demonstrate their utility in safe learning. Our model falls into the latter category, draws inspiration from the extensive literature on Pavlovian-instrumental interactions (*Brown and Jenkins, 1968*; *Kamin et al., 1963*; *Mackintosh, 1983*; *Maia, 2010*; *Mowrer, 1960*; *Mowrer, 1951*; *Prévost et al., 2012*; *Talmi et al., 2008*), fear conditioning (*LaBar et al., 1998*), and punishment-specific prediction errors (*Berg et al., 2014*; *Elfwing and*

*Seymour, 2017*; *Pessiglione et al., 2006*; *Roy et al., 2014*; *Seymour et al., 2007*; *Seymour et al., 2012*; *Watabe-Uchida and Uchida, 2018*), and elucidates a safety-efficiency trade-off. Classical theories of avoidance such as two-factor theory (*Mowrer, 1951*), and indeed actor-critic models (*Maia, 2010*), intrinsically invoke Pavlovian mechanisms in control, although primarily to act as a teaching signal for instrumental control as opposed to directly biasing action propensities such as in our case or (*Dayan et al., 2006*). Recent studies in computer science, particularly those employing policy optimisation (gradient-based) RL under CMDPs (*Altman, 1999*), have also observed a similar safety-efficiency trade-off (*Moskovitz et al., 2023*). Additionally, the fundamental trade-off demonstrated by *Fei et al., 2020*, between risk sensitivity (with exponential utility) and sample efficiency in positive rewards aligns with our perspective on the safety-efficiency trade-off, especially when broadening our definition of safety beyond cumulative pain to include risk considerations. Safety-efficiency trade-offs may also have a close relationship with maladaptive avoidance (*Ball and Gunaydin, 2022*) often measured in clinical anxiety, and our work provides insights into the maladaptive avoidance via the heightened threat appraisal pathway. Similar safe exploration behaviour in choices could be achievable using a risk-sensitive criterion such as conditional value at risk (CVaR) that relies only on the instrumental systems without needing a Pavlovian system. However, these work in different ways. Embracing the decision-theoretic psychiatry framework by *Huys et al., 2015*, which attempts to categorise dysfunctions as the agent either solving the problem with a wrong solution or solving the wrong problem correctly, or solving the right problem correctly but in an unfortunate or wrong environment, then we see the following. CVaR provides the correct solution to the wrong problem (an objective that only maximises the lower tail of the distribution of outcomes). In contrast, the Pavlovian bias provides the wrong solution to the correct problem (normative objective) (*Huys et al., 2015*). Further, approaches such as CVaR might not be the best approach to capture the Pavlovian withdrawal bias effect we find in RTs.

When it comes to our experiments, both the simulation and VR experiment models are related and derived from the same theoretical framework, maintaining an algebraic mapping. They differ only in task-specific adaptations, i.e., differ in action sets and differ in temporal difference learning rules - multi-step decisions in the grid world vs RW rule for single-step decisions in the VR task. This is also true for *Dayan et al., 2006*, who bridge Pavlovian bias in a Go-No Go task (negative auto-maintenance pecking task) and a grid world task. A further minor difference between the simulation and VR experiment models is the use of a baseline bias in the human experiment's RL and the RLDDM model, where we also model RTs with drift rates, which is not a behaviour often simulated in the grid world simulations. As mentioned previously, we use the grid world tasks for didactic purposes, similar to *Dayan et al., 2006*, and common to test-beds for algorithms in RL (*Sutton and Barto, 1998*). The main focus of our work is on Pavlovian fear bias in safe exploration and learning, rather than on its role in complex navigational decisions. Future work can focus on capturing more sophisticated safe behaviours, such as escapes (*Evans et al., 2019*; *Sporrer et al., 2023*) and model-based planning, which span different aspects of the threat-imminence continuum (*Mobbs et al., 2020*).

In our simulation experiments, we assume the coexistence of the Pavlovian fear system and the instrumental system to demonstrate the emergent safety-efficiency trade-off from their interaction. It is possible that similar behaviours could be modelled using an instrumental system alone, with higher punishment sensitivity; therefore, we do not argue for the necessity for the Pavlovian fear system here. Instead, the Pavlovian fear system itself could be a potential biologically plausible implementation of punishment sensitivity. Unlike punishment sensitivity (scaling of the punishments), which has not been robustly mapped to neural substrates in fMRI studies, the neural substrates for the Pavlovian fear system are well known (e.g. the limbic loop and amygdala, further see *Appendix 1—figure 11*). Additionally, the Pavlovian fear system provides a separate punishment memory that cannot be erased by greater rewards like (*Elfwing and Seymour, 2017*; *Wang et al., 2018*). This fundamental point can be observed in our simple T-maze simulations, where the Pavlovian fear system encourages avoidance behaviour and the agent chooses the smaller reward instead of the greater reward. We next discuss the plausibility of pre-training to select the hardwired actions. In the human experiment, the withdrawal action is straightforwardly biased, as noted, while in the grid world, we assume a hardwired encoding of withdrawal actions for each state/grid. This innate encoding of withdrawal actions could be represented in the dPAG (*Kim et al., 2013*). We implement this bias using pre-training, which we assume would be a product of evolution. Alternatively, this could be interpreted as deriving from

an appropriate value initialisation where the gradient over initialised values determines the action bias. Such aversive value initialisation, driving avoidance of novel and threatening stimuli, has been observed in the tail of the striatum in mice, which is hypothesised to function as a Pavlovian fear/threat learning system (*Menegas et al., 2018*).

We illustrate that a high Pavlovian impetus is characterised by reduced sample efficiency in learning, worsened/weakened (instrumental) value propagation and impervious rigidity in the policy, and misbehaviour due to misalignment of bias with the instrumental action. This way, it also further promotes short-term safer, smaller rewards opposed to long-term higher rewards, echoing the idea of Pavlovian pruning of decision trees (*Huys et al., 2012*). The idea of alignment between the Pavlovian and instrumental actions leading to harm-avoiding safe behaviours and misalignment being the root of maladaptive behaviours was proposed by *Mkrtchian et al., 2017b*, through a Go-No Go task with human subjects and the threat of shock responsible for the Pavlovian instrumental transfer. Recently, *Yamamori et al., 2023*, have developed a restless bandit-based approach-avoidance task to capture anxiety-related avoidance, by using the ratio of reward and punishment sensitivities as a computational measure of approach-avoidance conflict. We show in our simulations that misalignment can also lead to safe behaviours, but at the cost of efficiency. But having a flexible fear commissioning alleviates the majority of Pavlovian misbehaviour and in turn makes the agent more cautious in the face of uncertainty and catastrophe, contrasting with 'optimism bias' observed in humans (*Sharot, 2011*). A limitation of our work would be that we do not model the endogenous modulation of pain and stress-induced analgesia which may have the opposite effect of the proposed uncertainty-based fear commissioning scheme. A limitation of our VR experiment is that we only consider uncertainty decrease from first half to second half. This was motivated to make it similar to the grid world simulations as well as to help with behavioural tests (*Appendix 1—figure 5*), as this would keep all of the reducible and irreducible uncertainty in the first half and none in the second half. However, a stringent test would also require a balanced case, where the outcomes of cues 3 and 4 are more certain in the first half and more uncertain in the second half, or consider differentiating uncertainty and volatility.

While our flexible $\omega$ scheme, rooted in associability, shares motivation with *Dorfman and Gershman, 2019*, to track uncontrollability, our approach differs. Unlike *Dorfman and Gershman, 2019*, which employs a Bayesian arbitrator emphasising the most useful predictor (Pavlovian or instrumental), our Pearce-Hall associability-based measure provides a direct and separate controllability assessment. This distinction allows our measure to scale effectively to complex tasks, including grid world environments, maintaining stability throughout experiments. In contrast, the measure by *Dorfman and Gershman, 2019*, exhibits notable variability, even when the controllability of the cue-outcome pair remains constant throughout the task. Previous fMRI studies have associated associability signals with the amygdala (*Zhang et al., 2016*) and pgACC (*Zhang et al., 2018*) control. Additionally, outcome uncertainty computation could possibly be performed within the basal ganglia using scaled prediction errors (*Mikhael and Bogacz, 2016*; *Möller et al., 2022*) and is encoded in the firing rates of orbitofrontal cortex neurons and possibly in slow ramping activity in dopaminergic midbrain neurons (*Bach and Dolan, 2012*; *Fiorillo et al., 2003*; *O'Neill and Schultz, 2010*). Associability as a measure of outcome uncertainty, though very practical and useful on an implementational level, cannot distinguish between various kinds of uncertainties. Further, future work can help differentiate between controllability and predictability. *Ligneul et al., 2022*, suggest that controllability and not predictability may arbitrate the dominance of Pavlovian vs instrumental control. Future work could also use a formal account of uncertainty, which could fit the fear-conditioned skin-conductance response better than Pearce-Hall associability (*Tzovara et al., 2018*). Additionally, *Cavanagh et al., 2013*, demonstrated that theta-band oscillatory power in the frontal cortex tracks and overrides Pavlovian biases, later suggesting its connection to inferred controllability (*Gershman et al., 2021*). Notably, *Kim et al., 2023*, revealed that upregulation of the dorsolateral prefrontal cortex (dlPFC) through anodal transcranial direct current stimulation induces behavioural suppression or changes in Pavlovian bias in the punishment domain, implying a causal role of the dlPFC in Pavlovian-instrumental arbitration.

A natural clinical application (*Fullana et al., 2020*) of this model is towards mechanistic models of anxiety and chronic pain. Quite simply, both have been considered as reflecting excessive Pavlovian punishment learning systems. In the case of anxiety disorder, this equates a strong influence of Pavlovian control with subjectively experienced anxiety symptomatology, leading to excessively defensive behaviour and avoidance of anxiogenic environments (*Norton and Paulus, 2017*). In the

case of chronic pain, the idea is that failure to overcome a Pavlovian incentive to avoid moving results in failure to discover that pain escape is possible (the fear avoidance model) (*Vlaeyen and Linton, 2000*). In both cases, the pathological state can be considered a failure to turn down the Pavlovian system when the environment becomes more predictable (i.e. less uncertain). This illustrates a subtle distinction between existing theories that simply propose a constant excess Pavlovian influence, from the possibility they might result from a deficit in the flexible commission of Pavlovian control. This distinction can therefore be experimentally tested in clinical studies. Furthermore, accruing evidence also indicates a role of excessive Pavlovian punishment learning in models of depression (*Huys et al., 2016*; *Nord et al., 2018*), suggesting that this may be a common mechanistic factor in comorbidity between chronic pain, anxiety, and depression. Recent experiments and perspectives also suggest a psychological mechanism of how avoidance in humans can lead to growth of anxiety (increased belief of threats) (*Urcelay, 2024*). A key distinctive prediction of our model for an intervention is that we should help patient groups reduce Pavlovian bias not by training to reduce the bias, but rather by attempting to make the arbitration more flexible. This could potentially be done via some sort of controllability discrimination paradigm, i.e., helping distinguish between what is controllable and what is not - this is something also found in stoicism-based approaches to cognitive behavioural therapy (*Thorn and Dixon, 2007*; *Turk and Rudy, 1992*).

In conclusion, we outline how the Pavlovian fear system provides an important and computationally precise mechanism to shape or sculpt instrumental decision-making. This role for the Pavlovian fear system extends its utility far beyond merely representing the evolutionary vestiges of a primitive defence system, as sometimes portrayed. This opens avenues for future research in basic science of safe self-preserving behaviour (including in artificial systems), and clinical applications for mechanistic models of anxiety and chronic pain.

## Materials and methods

### Instrumental learning and Pavlovian fear learning

We consider a standard RL setting in an environment containing reward and punishments (pain). In each time step $t$, the agent observes a state $s_t$ and selects an action $a_t$ according to its stochastic policy $\pi_t(s_t, a_t)$ (i.e. the probability of selecting action $a_t = a$ in state $s_t = s$). The environment then makes a state transition from the current state $s_t$ to the next state $s_{t+1}$ and the agent receives a scalar reward $R_{t+1} \in (-\infty, +\infty)$. This represents that the scalar reward includes both positive rewards and negative rewards or punishments. We use the standard notation used by *Sutton, 2018*.

In the instrumental system, we define the value of taking action $a$ in state $s$ under a policy $\pi$, denoted as the action-value function $Q^\pi(s, a)$, as the expected return starting from $s$, taking the action $a$, and thereafter following policy $\pi$:

$$Q^\pi(s, a) = \mathbb{E}_\pi \left[ \sum_{k=0}^{\infty} \gamma^k R_{t+k+1} | s_t = s, a_t = a \right], \tag{1}$$

where $R_{t+k+1}$ is the scalar reward received $k$ time steps in the future, when evaluating the $Q$-values at time steps $t$. The discount factor is $\gamma$ and the reward $k$ time steps into the future is discounted by $\gamma^k$. The optimal action-value function is defined as $Q^*(s, a) = \max_\pi Q^\pi(s, a)$. Note that these are purely instrumental $Q$-values and do not include the Pavlovian bias.

In addition to the instrumental system, we define a Pavlovian fear (i.e. punishment/pain) system over and above the instrumental system which makes it safer. The Pavlovian fear systems aim to increase the impetus of the pain-avoidance actions that minimise pain. For that, we split the standard reward $R$ and only extract the punishment feedback signal $p \geq 0$:

$$p = -\min(R, 0), \tag{2}$$

We can similarly define a Pavlovian reward system trained on $max(R, 0)$, however, that's not relevant to the questions of this study, so we will only focus on the arbitration between the instrumental (state-action based) model-free and Pavlovian (state-based) fear system. And we define the pain state-value $V_p(s)$ of the Pavlovian fear system as follows:

$$V_p^\pi(s) = \mathbb{E}_\pi\left[\sum_{k=0}^\infty \gamma^k p_{t+k+l}|s_t = s\right], \tag{3}$$

The subset of actions with the Pavlovian bias $A_p$ are arrived at using a pre-training in the same environment with only punishments and random starting points. $V_p(s)$ then biases this pre-trained subset of actions $A_p$ according to *Equation 13*.

Here onwards, we will drop the time subscript for simplicity and write value update equations considering the state transitions from $s$ to $s'$. The Pavlovian fear state-value functions are updated as follows:

$$V_p(s) := V_p(s) + \alpha(p + \gamma V_p(s') - V_p(s)). \tag{4}$$

The instrumental value function for qualitative value plots is updated in an on-policy manner as follows (but is not used in the PAL algorithm):

$$V(s) := V(s) + \alpha(R + \gamma V(s') - V(s)). \tag{5}$$

And the instrumental action-value functions are updated as follows:

$$Q(s,a) := Q(s,a) + \alpha(\delta), \tag{6}$$

where $\alpha$ is the learning rate and while using off-policy Q-learning (sarsamax) algorithm, the TD-errors are calculated as follows:

$$\delta = R + \gamma Q(s', \mathrm{argmax}_{a'}(Q(s',a'))) - Q(s,a). \tag{7}$$

The equations above are valid for a general case and are used in grid world simulations. For model-fitting purposes for the VR approach-withdrawal task, there is no next state $s'$, thus the equations reduce to a simpler form of the RW learning rule.

## Action selection

Let $A$ be the action set. In the purely instrumental case, propensities $\rho(s,a)$ of actions $a \in A$ in state $s$ are the advantages of taking action $a$ in state $s$:

$$\rho(s,a) = Q(s,a). \tag{8}$$

And thus using softmax action selection with a Boltzmann distribution, the stochastic policy $\pi(a|s)$ (probability of taking action $a$ in state $s$) as follows:

$$\pi(a|s) = \frac{e^{(\rho(s,a)/\tau)}}{\sum_{a' \in A} e^{(\rho(s,a')/\tau)}}, \tag{9}$$

where $\tau$ is the temperature that controls the trade-off between exploration and exploitation. For grid world simulations, we use hyperbolic annealing of the temperature, where the temperature decreases after every episode $i$:

$$\tau(i) = \frac{\tau_0}{1 + \tau_k i}. \tag{10}$$

Here, $\tau_0$ is the initial temperature and $\tau_k$ controls the rate of annealing. This is to ensure the policy converges in large state spaces like a grid world and follows previous studies (*Elfwing and Seymour, 2017*; *Wang et al., 2018*). For model fitting of the VR approach-withdrawal task, we do not anneal it and keep it as a free parameter (inverse temperature $\beta = 1/\tau$) to be fitted to each participant. This is also consistent with previous literature (*Cavanagh et al., 2013*; *Dorfman and Gershman, 2019*; *Gershman et al., 2021*; *Guitart-Masip et al., 2012*) and several other works modelling Go-No Go tasks.

In the case of a Pavlovian pain/fear system, let $A_p$ be the subset of actions in state $s$ which has the Pavlovian pain urges or impetus associated with it. These are usually a small set of species-specific defensive reactions. In the VR approach-withdrawal task, we assume, or rather propose, it is the bias to withdraw from potentially harmful stimuli (in our case jellyfish). For the purpose of the grid world simulations, these can either be hardcoded geographical controller moving away from harmful states

or through a pre-trained value-based controller (*Dayan et al., 2006*). This work does not delve into the evolutionary acquisition of these biases, but one can derive the action subset $A_p$ from evolutionarily acquired value initialisations which may also help avoid novel stimuli and is a direction for future work.

Thus, after adding a Pavlovian fear system over and above the instrumental system, the propensities for actions are modified as follows:

$$\rho(s, a_n) = (1 - \omega)Q(s, a_n); \text{ where } a_n \in A_n = A \setminus A_p. \tag{11}$$

$$\rho(s, a_p) = (1 - \omega)Q(s, a_p) + \omega(V_p(s)); \text{ where } a_p \in A_p. \tag{12}$$

The same can be compactly written as mentioned in the illustration (*Figure 1*):

$$\rho(s, a) = (1 - \omega)Q(s, a) + \omega(V_p(s, a)); \text{ where } V_p(s, a) = \mathbb{I}[a = a_p]V_p(s). \tag{13}$$

where $\omega$ is the parameter responsible for Pavlovian-instrumental transfer. These equations are constructed following the preceding framework by *Dayan et al., 2006*, which laid out the foundation for interplay between Pavlovian reward system and the instrumental system. $\mathbb{I}[\cdot] = 1 \forall a_p \in A_p$ and $\mathbb{I}[\cdot] = 0 \forall a_n \in A_n = A \setminus A_p$ following the succinct vectorised notation by *Dorfman and Gershman, 2019*.

We have referred to this algorithm as the PAL algorithm in this paper. The equations above assume only a Pavlovian fear system in addition to an instrumental system, and the given equations would vary depending on if we add a Pavlovian reward system too. After this modification, the action selection probabilities are calculated in a similar fashion as described in *Equation 9*.

## Uncertainty-based modulation of $\omega$

We further modulate the parameter $\omega$ which is responsible for Pavlovian-instrumental transfer using perceived uncertainty in rewards. We use Pearce-Hall associability for this uncertainty estimation based on unsigned prediction errors (*Krugel et al., 2009*; *Zhang et al., 2016*; *Zhang et al., 2018*). We maintain a running average of absolute TD-errors $\delta$ (*Equation 7*) at each state using the following update rule:

$$\Omega_{t+1} = (1 - \alpha_\Omega * \alpha)\Omega_t + \alpha_\Omega * \alpha|\delta|. \tag{14}$$

where $\Omega$ is the absolute TD-error estimator, $\alpha$ is the learning rate for $Q(s, a)$, and $V(s)$ values as mentioned earlier and $\alpha_\Omega \in [0, 1]$ is the scalar multiplier for the learning rate used for running average of TD-error. To obtain parameter $\omega \in [0, 1]$ from this absolute TD-error estimator $\Omega \in [0, \infty)$, we scale it linearly using scalar $\kappa$ and clip it between [0,1] as follows:

$$\omega_t = \min(\kappa\Omega_t, 1). \tag{15}$$

We note that the range values $\Omega$ takes largely depends on the underlying reward function in the environment and $\alpha_\Omega$. Thus, we choose a suitable value of $\kappa$ for $\alpha_\Omega$ using grid search in each grid world environment simulation to ensure that the Pavlovian system dominates in cases of high uncertainty and that the instrumental system starts to take control as uncertainty reduces. We aim to show that this flexible $\omega$ scheme is a viable candidate for arbitration between the two systems and addresses the safety-efficiency dilemma wherever it arises. The initial associability $\Omega_0$ is set to 0 in grid world simulations as there is no principled way to set it. In the case of model fitting for the VR approach-withdrawal task, $\Omega_0$, $\kappa$, and $\alpha_\Omega$ are set as free parameters fitted to each participant and instead of the TD-errors, we have the RW rule equivalent - punishment prediction errors without any next state $s'$.

## Grid world simulation details

We consider a series of painful grid world-based navigational tasks, including moderate sources of pain (more variations in the Appendix with catastrophic and dynamic sources of pain). In the grid worlds, the goal is to navigate from starting position (in blue) to goal position (in green) while avoiding the static moderately painful states (in red). The agent receives a positive reward of 1 for reaching the goal and pain of 0.1 for moderately painful states (red). The pain is encoded as a negative reward of –0.1 in the case of standard RL. Four actions move the agent one step north, south, east, or west (or choose not to move, allowed only in certain environments). If the agent hits a wall, it stays and remains in its current state. All simulation environments have the following stochastic transition probabilities: 0.9

probability of correct (desired) state transition, whereas with 0.05 probability, the agent's state transitions to the position perpendicular to action taken (right or left). We test the PAL algorithm for varying $\omega = 0.1, 0.5, 0.9$ and for uncertainty-based modulation of flexible $\omega$ and compare the performance with standard instrumental policy ($Q$-learning). The following meta parameters are fixed for all our tabular grid world simulations - learning rate $\alpha = 0.1$, discount factor $\gamma = 0.99$, temperature annealing meta parameters $\tau_0 = 1, \tau_k = 0.025$. The meta parameters $\alpha_\Omega$ and $\kappa$ are tuned using grid search on the safety-efficiency trade-off metrics for each environment. This is necessary, as different environments have different underlying reward distributions leading to different distributions of TD-errors, thus its running average needs to be appropriately scaled to map it to $\omega \in [0, 1]$. Due to this meta-parameter tuning, the claim in the simulation experiments is a modest one that there exists a $\alpha_\Omega$ and $\kappa$ that mitigate the trade-off as opposed to the trade-off being mitigated by every possible combination of $\alpha_\Omega$ and $\kappa$. This resembles the model-fitting procedure in other experimental tasks, where $\alpha_\Omega$ and $\kappa$ are fit in a hierarchical Bayesian manner, suggesting that humans perform this tuning to varying degrees to the best of their ability. The $Q$-tables and $V_p$-tables are initialised with zeros. Plots are averaged over 10 runs with different seed values.

We quantify safety using cumulative pain accrued by the agent, and sample efficiency using the cumulative steps (or environment interactions or samples) taken by the agent across all the episodes in the learning process. The lesser the cumulative pain accrued over episodes, the safer is the learning; and the lesser the cumulative steps (or environment interactions), the more efficient is the learning in terms of reward seeking and task completion in each episode. Furthermore, we also construct a trade-off metric to measure how well the safety-efficiency trade-off is improved. We define the safety-efficiency trade-off metrics as follows, which is maximised when both cumulative pain and cumulative steps are independently minimised:

$$\text{Trade-off metric} = \frac{1}{CP_n^2 + CS_n^2},\tag{16}$$

where $CP_n$ and $CS_n$ are cumulative pain and cumulative steps normalised by dividing the maximum cumulative pain and steps achieved (usually by fixed $\omega = 0$ or $\omega = 0.9$) in that run. We acknowledge that this normalisation can make the metric favour improvements in either safety or efficiency unequally to an extent, as it weighs the improvements in safety or efficiency relative to worst performance in each of them. This metric can be further weight-adjusted to give more priority to either CP or CS as required, but we don't do that in the current instance. Thus, this metric should only be used as a didactic tool and not an absolute metric of performance, and one should instead draw conclusions by observing the cumulative pain accrued and steps taken over multiple episodes.

## Approach-withdrawal conditioning task: experimental design
### Participant recruitment and process
30 adults participated in the experiment (15 females, 15 males; age: min=18, max=60, mean=30.5, standard deviation=12.44). Healthy participants from ages 18–60 were allowed to participate in the study (pre-established inclusion criteria). All subjects provided written informed consent for the experiment, which was approved by the local ethics board - University of Oxford Central University Research Ethics Committee (CUREC2 R58778/RE002). One participant withdrew and did not complete the study, and one participant turned out to be a fibromyalgia patient upon arrival, thus was excluded. The rest of the 28 healthy subjects' (14 female, average age 27.96 years) data was used for the analysis.

Participants filled a short demographic form upon arrival, followed by a pain tolerance calibration procedure, followed by putting on all of the sensors, followed by a re-calibration of their pain tolerance before starting the practice session and the main experiment. All of this was usually completed within 2 hr and participants were paid £30 for their participation (and were adequately compensated for any unexpected overtime and reasonable travel reimbursements). They were free to withdraw from the experiment at any time.

### Trial protocol
We use a trial-based approach with a withdrawal task; however, the subjects had complete control over when to start the next trial. Each trial consisted of four events: a choice to initiate the trial, a coloured jellyfish cue, an approach or withdrawal motor response, and a probabilistic outcome.

The timeline is displayed in *Figure 1*. In each trial, subjects will initiate the trial by bringing in their hand inside a hovering bubble in front of them. Then a jellyfish will emerge and fade in (gradually decreasing transparency) within the next 0.5 s and then stay in front of the subject for another 1 s, making the total fixation segment 1.5 s long. Throughout the fixation segment, the jellyfish colour will remain greyish-black. After this fixation segment terminates, the jellyfish takes one of the four colours with associated pain outcome contingencies. This is the stimulus phase and the subject is required to perform either an approach or a withdrawal response within the next 2 s. The approach response involved reaching out their hand and touching the jellyfish, whereas the withdrawal response involved withdrawing the hand away from the jellyfish and towards one's own self. The subjects practised these two actions in the practice session before the main experiment and were instructed to perform either of these two actions. The stimulus ended as soon as an action was successfully completed and was followed by the probabilistic outcome phase. In the rare case that the 2 s time window completed before the subject could successfully perform either of these two actions, then for the purpose of the probabilistic outcome segment, the action was decided based on the hand-distance from the jellyfish (i.e. whether it was closer to an approach or a withdrawal action). The possible outcomes were either a painful electric shock (along with some shock animation visualisations around the jellyfish) or a neutral outcome (along with bubble animations from the jellyfish). The outcomes were presented depending on the action taken and the contingencies for each cue, as shown in *Figure 4*. After the outcome segment which lasted for 1.5 s, the jellyfish proceeded to fade out (become more transparent gradually) for the next 0.75 s and then the subject could start the next trial by again bringing their hand within the bubble in front of them.

Subjects were instructed to try to keep their hand inside the bubble during this fixation segment and only move the hand after the jellyfish changes colour. The bubble was placed halfway between the subject and the jellyfish and was placed slightly to the right for right-handed and slightly to the left for left-handed subjects. The subjects performed the task with their dominant hand.

## Block protocol

Prior to the main task and the practice session, we perform a calibration of the intensity of pain stimulation used for the experiment according to each individual's pain tolerance. To do this, we start with the minimum stimulation value and gradually increase the value using the 'staircase' procedure. We will record a 'threshold' value (typically rated as 3/10 on Likert scale), which is identified as the participant first reports pain sensation. We will record a second 'maximum' value, which the participant reports as the maximum pain sensation that the participant would be comfortable to tolerate for the complete experiment (typically rated 8/10 on the Likert scale). We then use 80% of that maximum value for stimulation throughout the experiment.

Before the main task, the subjects had to go through a short practice session to get acquainted with approach and withdrawal motions and the speed requirements. Subjects had one attempt at each of the two actions with no painful outcomes and no timeouts followed by a short practice session with two jellyfish (five trials each, randomised) and with 80% painful outcome contingencies for approach and withdrawal, respectively. They were informed as to which of these two jellyfish likes to be touched and which does not, during the practice session but not for the main experiment. The colours of the jellyfish for the practice session were different from those used for the main experiment. The four colours of the jellyfish cues for the main experiment were chosen so as to be colourblind-friendly. The main experiment had a total of 240 trials, 60 trials for each of the four jellyfish which was balanced across each quarter of the block, i.e., 15 trials per jellyfish per quarter block. The jellyfish 1 was the approach-to-avoid type and jellyfish 2 was the withdraw-to-avoid type throughout 240 trials. The jellyfish 3 was uncontrollable for the first half of the block (first 120 trials) and then was approach-to-avoid type for the rest of the block. The jellyfish 4 was uncontrollable for the first half of the block (first 120 trials) and then was the withdraw-to-avoid type for the rest of the block. Approach-to-avoid type means that the outcome would be a neutral outcome 80% of the times (and shock, 20% of the times) if the 'correct' approach action was performed, or else the outcome would be a shock 80% of the times (and neutral, 20% of the times) if the 'incorrect' withdrawal action was performed. Withdraw-to-avoid type means that the outcome would be a neutral outcome 80% of the times (and shock, 20% of the times) if the 'correct' withdraw action was performed, or else the outcome would be a shock 80% of the times (and neutral, 20% of the times) if the 'incorrect' approach action was performed.

Uncontrollable type means that the outcome could be shock or neutral with 50% probability each, regardless of the actions performed. After each quarter of the block, the subjects were informed of their progress through the block with a 10 s rest.

### Analysis

The choices and RTs were extracted and used for model fitting. The EEG, electromyography (EMG) and skin conductance data were acquired but found to be too corrupted by movement artefacts and noise to allow reliable analysis.

## Hierarchical Bayesian model-fitting choices and RTs

For both RL model fitting to choices and RLDDM model fitting to choices and RTs, we built four models each: RW (i.e. RW learning rule model), RW+bias (i.e. RW model with baseline bias), RW+bias+Pavlovian (fixed) (i.e. RW+bias model with a fixed Pavlovian withdrawal bias) and RW+bias+Pavlovian (flexible) (i.e. RW+bias model with a flexible Pavlovian withdrawal bias) similar to *Guitart-Masip et al., 2012*.

The action selection for RL models was performed using a softmax as per *Equation 9* with free parameter $\beta = 1/\tau$ and $\beta > 0$. The learning rule for RW models was:

$$Q(s,a) := Q(s,a) + \alpha(R - Q(s,a)), \tag{17}$$

where $R = -1$ in case of electric shocks or $R = 0$ in case of neutral outcome. Punishment $p$ can be defined as per *Equation 2* and thus $p = 1$ in case of electric shocks or $p = 0$, and the Pavlovian punishment value is calculated as per:

$$V_p(s) := V_p(s) + \alpha(p - V_p(s)) \tag{18}$$

$\alpha > 0$ is the learning rate and fitted as a free parameter and note that here $V_p$ is always positive.
For RW+bias model,

$$\rho(s,a) = Q(s,a) + b; \text{ if } a = \text{Approach} \tag{19}$$
$$\rho(s,a) = Q(s,a); \text{ else.} \tag{20}$$

Here, $b \in (-\infty, +\infty)$ is the baseline bias, which if positive represents a baseline approach bias and if negative represents baseline negative bias and is not Pavlovian in nature.
For RW+bias+Pavlovian (fixed) and RW+bias+Pavlovian (flexible) models,

$$\rho(s,a) = (1-\omega)Q(s,a) + b; \text{ if } a = \text{Approach} \tag{21}$$
$$\rho(s,a) = (1-\omega)Q(s,a) + \omega(V_p(s)); \text{ if } a = \text{Withdrawal} \tag{22}$$

Here, $\omega \in [0, 1]$ is a free parameter for the RW+bias+Pavlovian (fixed) model. $\omega$ is not a free parameter for the RW+bias+Pavlovian (flexible) model, but computed as per *Equations 14 and 15* with free parameters $\Omega_0$ (initial associability), $\kappa$ (scaling factor for $\omega$), and $\alpha_\Omega$ (learning rate multiplier for associability).

For RLDDM models, it is assumed that within a trial, the evidence is accumulated using a drift-diffusion process with parameters drift rate ($v$), non-decision time (ndt), threshold, and starting point. Non-decision time and threshold were kept as free parameters, and the starting point was kept constant and equal to half the threshold (making it equally likely starting point for both approach and avoidance actions). The drift rate $v$ was set according to the difference in action propensities between the choices as follows:

$$v = \rho(s, a = \text{Approach}) - \rho(s, a = \text{Withdrawal}). \tag{23}$$

Thus, the baseline bias and the Pavlovian biases were also included in the drift rate.
For model fitting, we used a hierarchical Bayesian modelling approach; all models were fit using Stan. They were fit using both custom code in PyStan, as well as using the hBayesDM package (*Ahn et al., 2017*), and final plots of group-level and subject-level parameter distributions were generated using the plotting functions in hBayesDM. Four parallel chains were run for all models. To assess the predictive accuracy of the models, we computed the LOOIC and WAIC (*Vehtari et al., 2017*).

## Software and hardware setup

We used the HTC Vive Pro Eye for the virtual reality (VR) with Alienware PC setup and the experiment was designed in Unity (game engine). The pain stimulator used was DS5 with WASP electrodes for the VR approach-withdrawal task and Silver-Silver Chloride (Ag/AgCl) Cup Electrodes for the VR maze task. We also collected galvanic skin response, heart rate (HR), EMG signals, wireless EEG using Brainproducts LiveAmp and Vive tracker movement signals and eye-tracking inside the VR headset.

The pain stimulator electrodes were attached to the ring finger, between the ring and the middle finger. The GSR sensors were attached to the middle and the index fingers, and the EMG sensors were attached to the brachioradialis muscle of the active hand used in the task with the ground electrode on the elbow. The HR sensor was attached to the index finger of the opposite hand.

## Acknowledgements

PM would like to thank Michael Browning, Rafal Bogacz, Suyi Zhang, Charlie Yan, Maryna Alves Rosa Reges, Danielle Hewitt, Katja Wiech, the anonymous COSYNE 2022 reviewers, CCN 2023 reviewers, and Science Advances reviewers for their feedback on earlier draft(s) of the subsections/extended abstracts of the manuscript. PM would like to thank Simon Desch for feedback on RLDDM fitting and Danielle Hewitt for suggestions and guidance on EEG data analysis. The work was funded by Wellcome Trust (214251/Z/18/Z, 203139/Z/16/Z and 203139/A/16/Z), IITP (MSIT 2019-0-01371), and JSPS (22H04998). This research was also partly supported by the NIHR Oxford Health Biomedical Research Centre (NIHR203316). The views expressed are those of the author(s) and not necessarily those of the NIHR or the Department of Health and Social Care. For the purpose of open access, the authors have applied a CC BY public copyright licence to any Author Accepted Manuscript version arising from this submission.

## Additional information

### Funding

| Funder | Grant reference number | Author |
| --- | --- | --- |
| Wellcome Trust | 10.35802/214251 | Ben Seymour |
| Wellcome Trust | 10.35802/203139 | Pranav Mahajan<br>Ben Seymour |
| Institute for Information and Communications Technology Promotion | MSIT 2019-0-01371 | Sang Wan Lee<br>Ben Seymour |
| Japan Society for the Promotion of Science | 22H04998 | Ben Seymour |
| NIHR Oxford Health Biomedical Research Centre | NIHR203316 | Ben Seymour |

The funders had no role in study design, data collection and interpretation, or the decision to submit the work for publication. For the purpose of Open Access, the authors have applied a CC BY public copyright license to any Author Accepted Manuscript version arising from this submission.

### Author contributions

Pranav Mahajan, Conceptualization, Data curation, Software, Formal analysis, Investigation, Visualization, Methodology, Writing – original draft, Writing – review and editing; Shuangyi Tong, Software; Sang Wan Lee, Funding acquisition, Writing – review and editing; Ben Seymour, Conceptualization, Supervision, Funding acquisition, Writing – review and editing

### Author ORCIDs

Pranav Mahajan https://orcid.org/0009-0001-2507-5450
Shuangyi Tong https://orcid.org/0009-0003-4985-6600

Sang Wan Lee https://orcid.org/0000-0001-6266-9613
Ben Seymour https://orcid.org/0000-0003-1724-5832

### Ethics

Human subjects: All human subjects in the study provided written informed consent for the experiment, which was approved by the local ethics board - University of Oxford Central University Research Ethics Committee (CUREC2 R58778/RE002).

Reviewer #1 (Public review): https://doi.org/10.7554/eLife.101371.3.sa1
Reviewer #2 (Public review): https://doi.org/10.7554/eLife.101371.3.sa2
Reviewer #3 (Public review): https://doi.org/10.7554/eLife.101371.3.sa3
Author response https://doi.org/10.7554/eLife.101371.3.sa4

---

## Additional files

### Supplementary files
MDAR checklist

### Data availability
The code and data used for generating the results are provided in GitHub (copy archived at *Mahajan, 2025*).

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

## Appendix 1

### Robustness of the associability-based $\omega$ in gridworld simulations

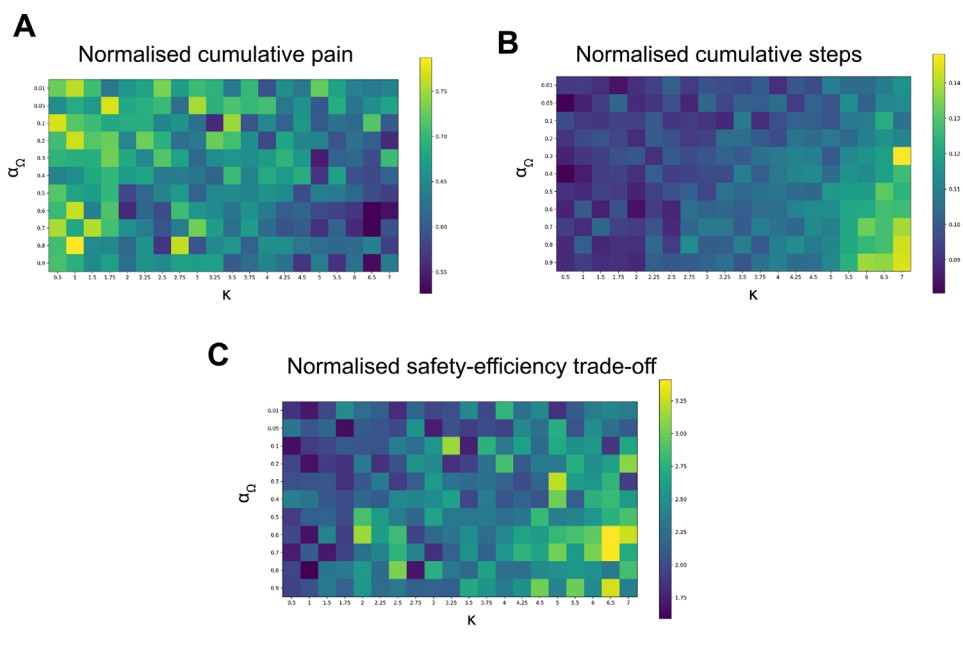

Meta parameters: $\kappa = 6$, $\alpha_\Omega = 0.4$

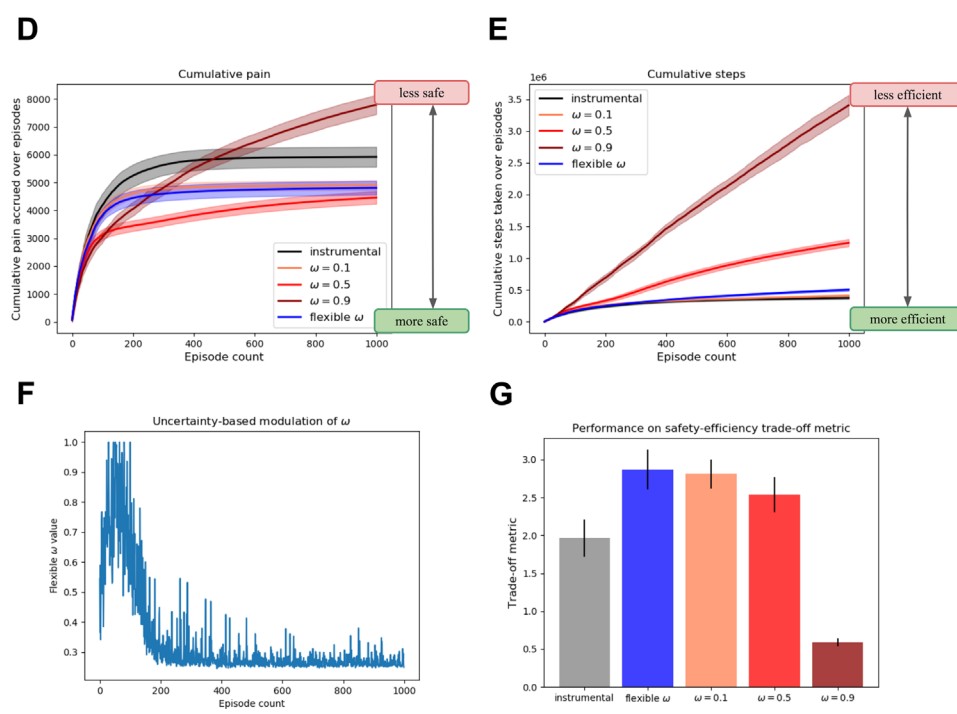

**Appendix 1—figure 1.** Robustness of PAL in gridworld simulation. This figure shows the robustness of grid search for tuning the meta parameters for the associability-based $\omega$ in grid world simulations. We show that the results hold for a range of values close to the chosen meta-parameters. (**A–C**) Grid search results for the environment in *Figure 2* for varying $\kappa$ and $\alpha_\Omega$. (**D–G**) Results for another set of meta-parameters.

# Flexible $\omega$ agent better adapts to reward relocation than a fixed $\omega$ agent

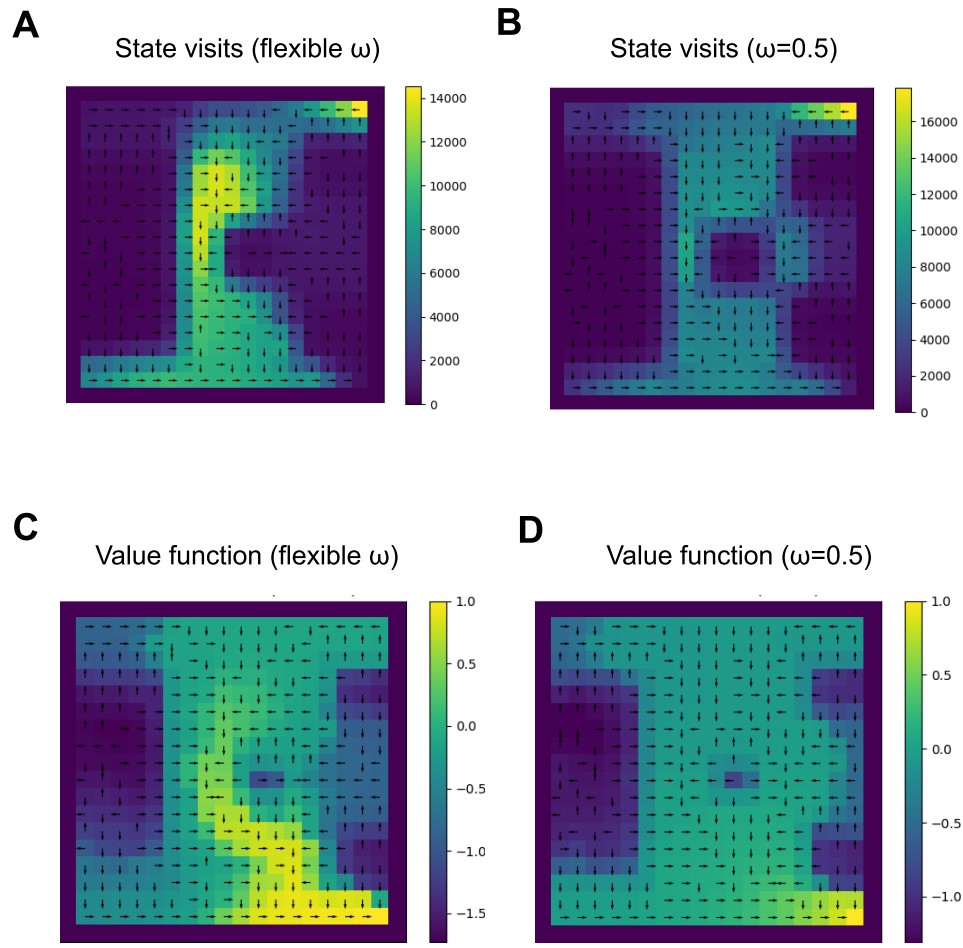

**Appendix 1—figure 2.** Additional reward relocation experiments with PAL. This figure shows cumulative state visit plots and value function plots of the flexible $\omega$ and fixed $\omega$ agents at the end of 1000 episodes when we relocate the reward goal from the bottom-left corner (**Figure 2**) to the bottom-right corner on episode 500. Comparing state visit plots (**A, B**) and comparing value function plots (**C, D**), we observe that persistent Pavlovian influence leads to persistent rigidity, while the flexible fear commissioning scheme is able to efficiently locate the goal. We observe that, unlike flexible $\omega$, constant $\omega$=0.5 leads to diminished value propagation of the rewarding value (**C, D**).

# Solving the safety-efficiency trade-off in a range of grid world environments

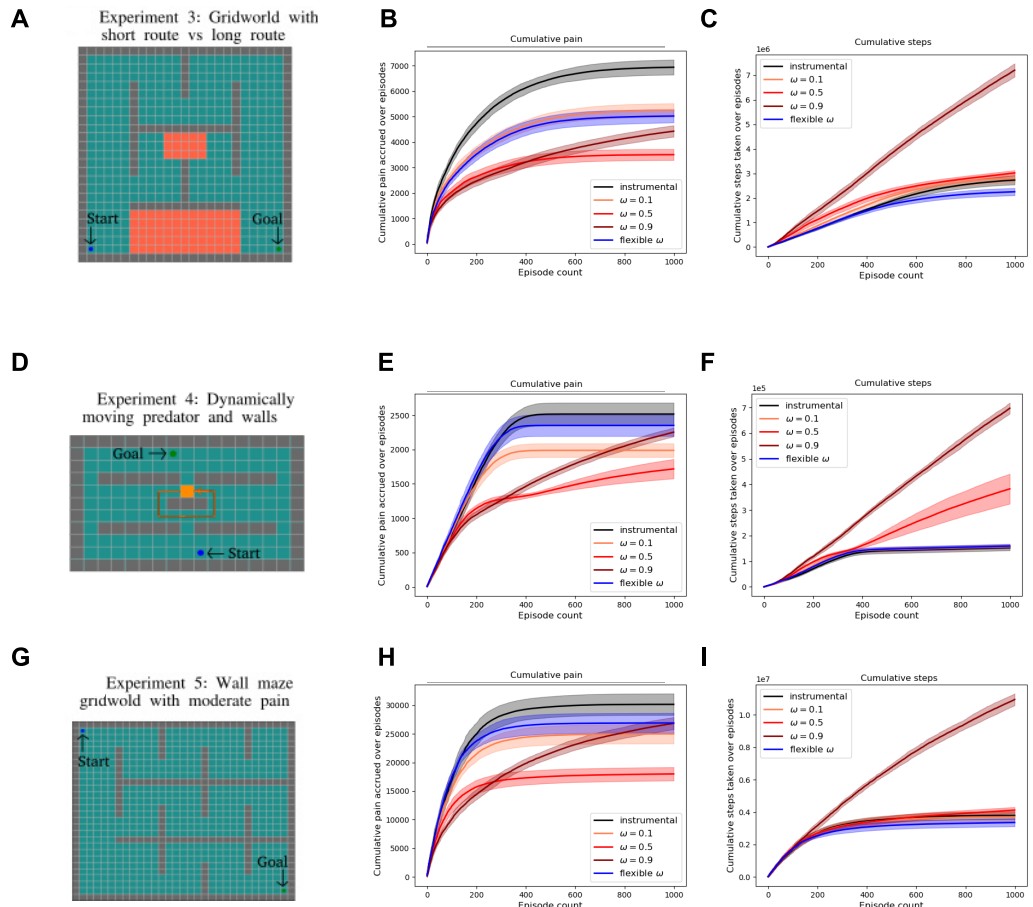

**Appendix 1—figure 3.** Flexible arbitration solves safety-efficiency trade-off in a range of gridworlds. In this figure, we show the performance of fixed $\omega$=0.1,0.5,0.9 and flexible $\omega$ agents on a range of grid world environments, namely (**A**) the three-route environment from *Figure 3*, (**D**) an environment with a moving predator on routine path and (**G**) wall maze grid world from *Elfwing and Seymour, 2017*. Colliding with the predator results in a negative reward of –1 and catastrophic death (episode terminates). Otherwise, colliding with the walls results in moderate pain of 0.1, and the agent's state remains unchanged. The latter two are completely deterministic environments, unlike the previous environments in the main paper. (**B, C, E, F, H and I**) show the safety-efficiency trade-off that arises in these three environments as well, and there is a separate optimal fixed $\omega$ for each environment. Alternatively, there exists a flexible $\omega$ scheme for each environment that can solve the trade-off, suggesting that the brain may be calibrating $\omega$ flexibly.

## Human three-route virtual reality maze results

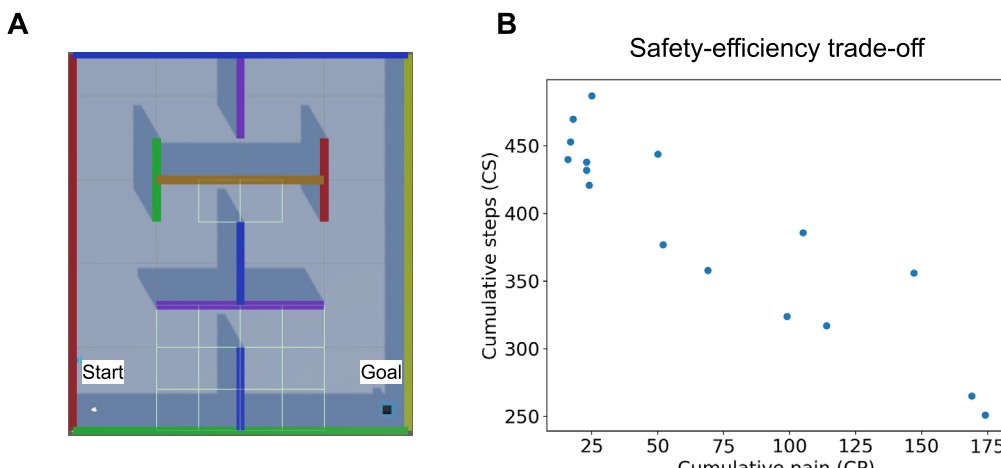

**Appendix 1—figure 4.** Additional results on human three-route VR maze task. (**A**) Top-view of virtual reality (VR) maze with painful regions annotated by highlighted borders. (**B**) Cumulative steps required to reach the goal vs cumulative pain acquired by participants over 20 episodes in the VR maze task. In this figure, we show the results of a VR maze replicating the three-route grid world environment from simulation results; however, it had fewer states and the participants were instructed to reach the goal which was visible to them as a black cube with 'GOAL' written on it. In order to move inside the maze, participants had to physically rotate in the direction they wanted to move and then press a button on the joystick to move forward in the virtual space. Thus, the participant did not actually walk in the physical space but did rotate up to 360 degrees in physical space. The painful regions were not known to the participants, but they were aware that some regions of the maze may give them painful shocks with some unknown probability. Walking over the painful states in the VR maze, demarcated by grid borders (see A), potentially shocked them with 75% probability while ensuring 2 s of pain-free interval between two consecutive shocks. Participants were not given shocks with 100% probability as that would be too painful for participants due to the temporal summation effects of pain. The participants engaged in 20 episodes of trials and were aware of this before starting the task and were free to withdraw from the experiment at any point. 16 participants (11 female, average age 30.25 years) were recruited and were compensated adequately for their time. The pain tolerance was acquired similarly to the approach-withdrawal task. (**B**) All participant trajectories inside the maze were discretised into an 8×9 (horizontal × vertical) grid. Entering a 1×1 grid section counted incremented the cumulative steps (CS) count. Upon receiving the shocks, the cumulative pain (CP) count was incremented. CP and CS over 20 episodes were plotted against each other to observe the trade-off. A limitation of this experiment is that it reflects the constraints of the grid world, and future experiments are necessary to show the trade-off in a range of environments.

## Behavioural results from approach-withdrawal VR task

We observe a baseline approach bias through significant asymmetry in average number of cumulative approaches and withdrawals (*Appendix 1—figure 5A*) and we consider the few inactive approaches and inactive withdrawals due to timeout as approaches and withdrawals, respectively (*Appendix 1—figure 5B*).

We consider a couple of model-free metrics of Pavlovian withdrawal bias prior to model fitting. The withdrawal bias metric on choices for two cues (say, cue X and cue Y) is calculated as follows:

$$\text{Choice bias metric(cue X, cue Y)} = \%\text{withdrawal choices on 'cue X'} - \%\text{approach choices on 'cue Y'} \quad (24)$$

and the metric for withdrawal bias in RTs is simply the subtraction of average withdrawal times from average approach times in a half (60 trials) or the quarter block (30 trials) under consideration. This choice metric is an extension of the metric used by *Dorfman and Gershman, 2019*, to punishment bias, and its logic is as follows. Consider the choice bias metric (cue2, cue1) - As Pavlovian withdrawals will increase %withdrawal choices i.e., (correct choice) for cue2 and decrease %approach choices, i.e., (correct choice) for cue1. Similarly, it will also make sense for metric (cue4, cue3) in the second half, albeit the bias would be lesser as they will be exploiting the optimal actions. It makes less

sense for (cue4, cue3) in the first half as there is no optimal action; however, it helps act as a control and quantify a baseline approach bias. Unfortunately, this metric cannot differentiate an action due to random exploration from an action due to Pavlovian misbehaviour, leading to noisy estimates. Further, it cannot capture baseline approach bias $b$ at all, because the model by *Dorfman and Gershman, 2019*, does not consider this parameter, unlike (*Cavanagh et al., 2013*; *Guitart-Masip et al., 2012*). However, we show that including baseline bias contributes the most to an incremental improvement in model fit.

We expect this bias to be largest in the first half with uncontrollable cues, and especially in the second quarter as opposed to the first quarter, by when enough punishment value would have been accrued for each of the cues and there would be a significant drop in random exploration. The Pavlovian withdrawal bias in choices is measured using the controllable cues 1 and 2 and thus computing the same quantity in uncontrollable cues 3 and 4 acts as control (*Appendix 1—figure 5C*). Likewise, we also hypothesised that there would exist a Pavlovian bias in RTs which speed up all withdrawals and slow down all approaches regardless of the cue (*Appendix 1—figure 5E*). For our second hypothesis, the Pavlovian bias should decrease with a decrease in outcome uncertainty, i.e., it would be higher in the second quarter as opposed to the fourth quarter (*Appendix 1—figure 5D and F*). We compare the quarters rather than the first and second halves to minimise the noise through random exploration in the first quarter. However, the differences we observe are not statistically significant.

In addition to these results from behavioural metrics in choices and RTs, we further observe certain change-of-mind-like patterns in motor responses. It is unclear if these are due to a Pavlovian bias or due to other factors and can be investigated in future studies.

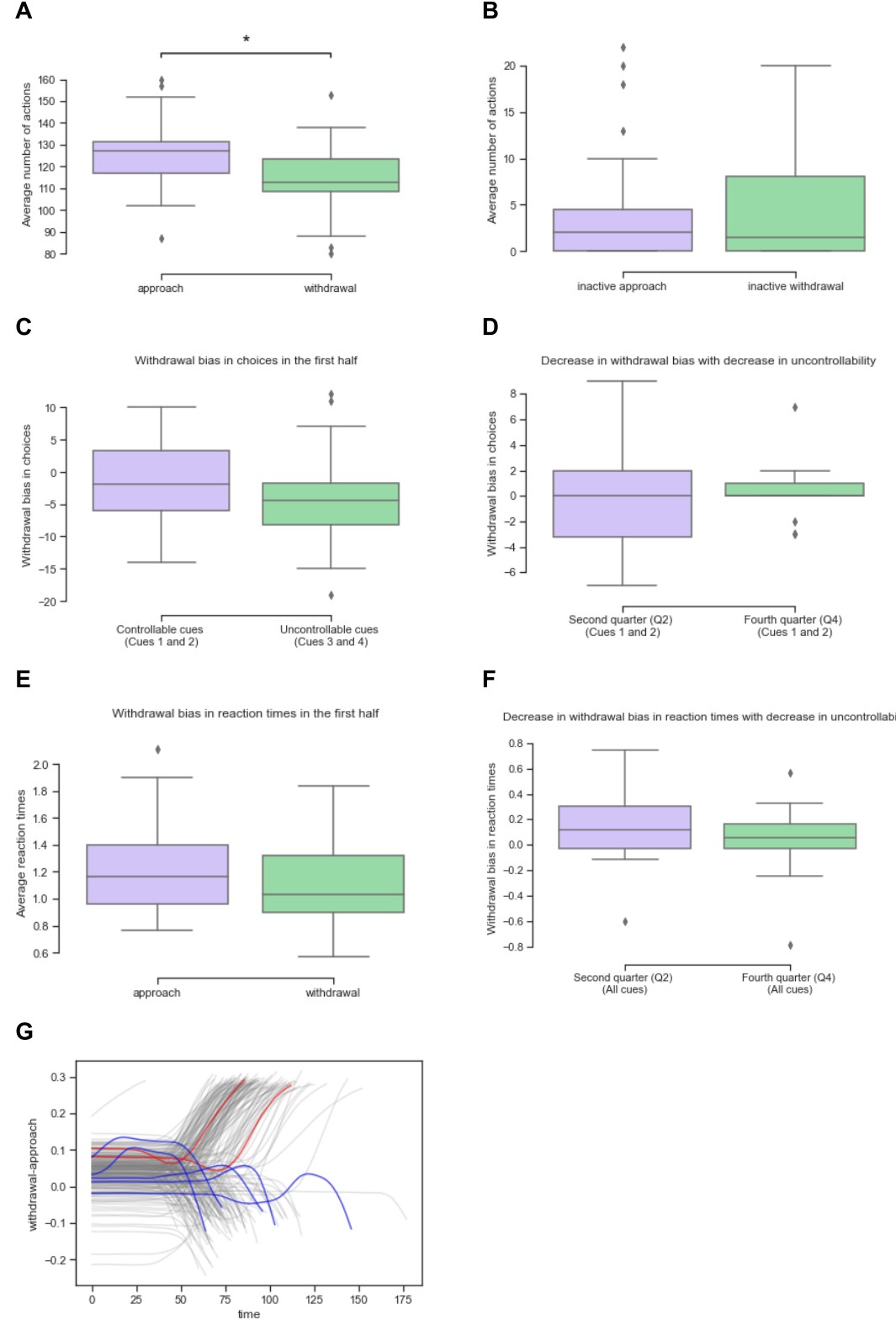

**Appendix 1—figure 5.** Behavioural results from Approach-Withdrawal VR task. (**A**) Asymmetry in average approach and withdrawal responses over all subjects showing a baseline approach bias - Mann-Whitney U test
*Appendix 1—figure 5 continued on next page*

*Appendix 1—figure 5 continued*

(statistic=597.0, p-value=0.0004). (**B**) Incomplete approaches and incomplete withdrawals that were counted as approaches and withdrawals, respectively. (**C**) Withdrawal bias in choices in the first half with uncontrollable cues - Mann-Whitney U test (statistic = 492.5, p-value = 0.0504). (**D**) Decrease in withdrawal bias in choice with decrease in uncontrollability - Mann-Whitney U test (statistic = 350.5, p-value = 0.7611). (**E**) Withdrawal bias in reaction times in the first half with uncontrollable cues - Mann-Whitney U test (statistic = 475.0, p-value = 0.0882). (**F**) Decrease in withdrawal bias in reaction times with decrease in uncontrollability - Mann-Whitney U test (statistic = 456.0, p-value = 0.1490). (**G**) Change-of-mind trials observed in motor data.

## Group and subject-level parameter distributions of RL and RLDDM models

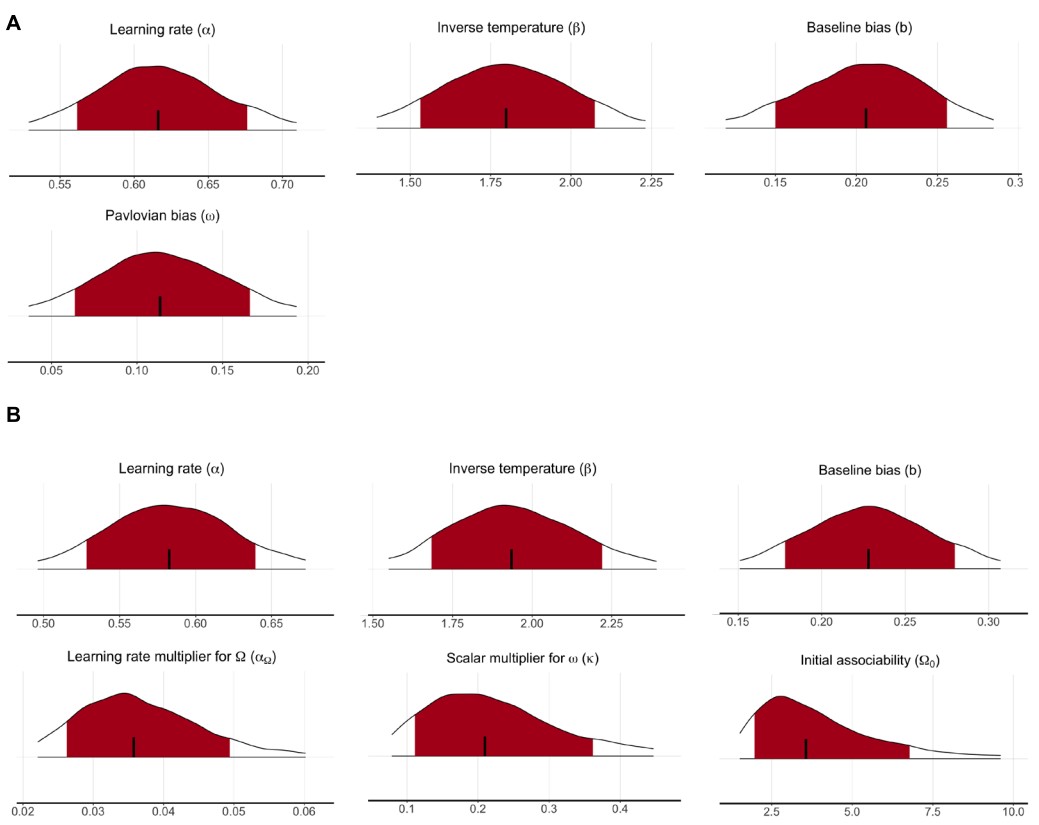

**Appendix 1—figure 6.** Group-level parameter distributions from RL models. Group-level parameter distributions from (**A**) the reinforcement learning (RL) model (M3) with fixed $\omega$ and (**B**) the RL model (M4) with flexible $\omega$. Shaded red regions denote 95% confidence intervals.

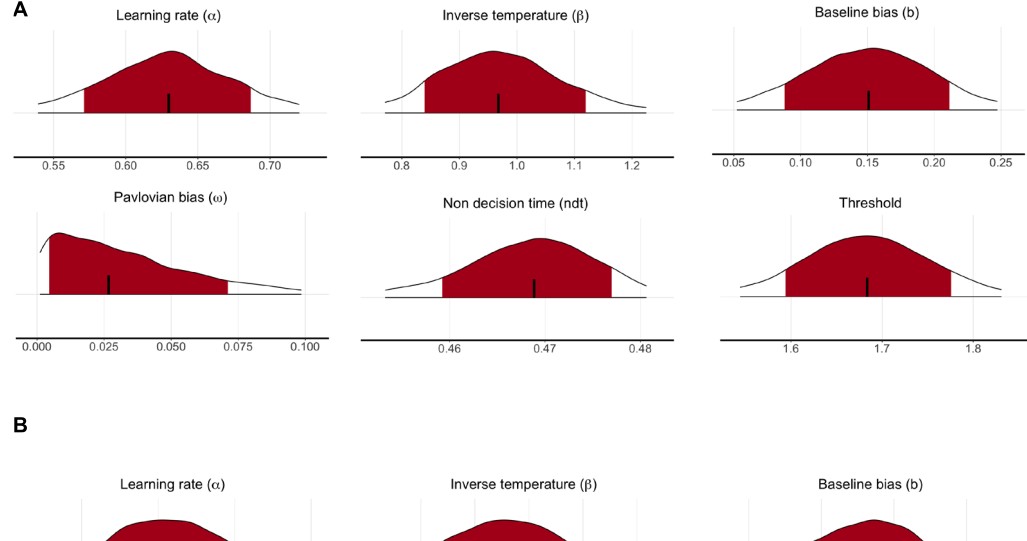

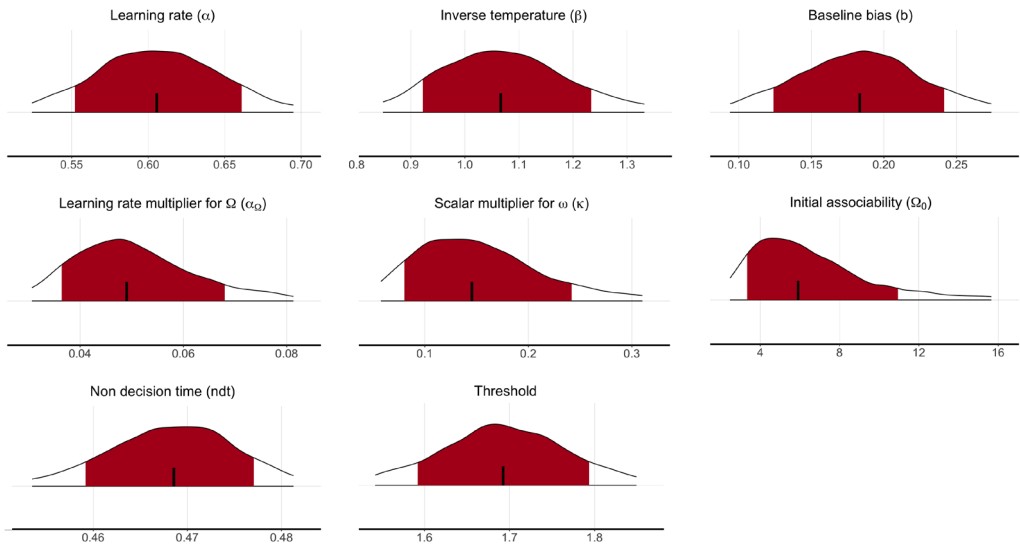

**Appendix 1—figure 7.** Group-level parameter distributions from RLDDM models. Group-level parameter distributions from (**A**) the reinforcement learning diffusion decision-making (RLDDM) model (M3) with fixed $\omega$ and (**B**) the RLDDM model (M4) with flexible $\omega$. Shaded red regions denote 95% confidence intervals.

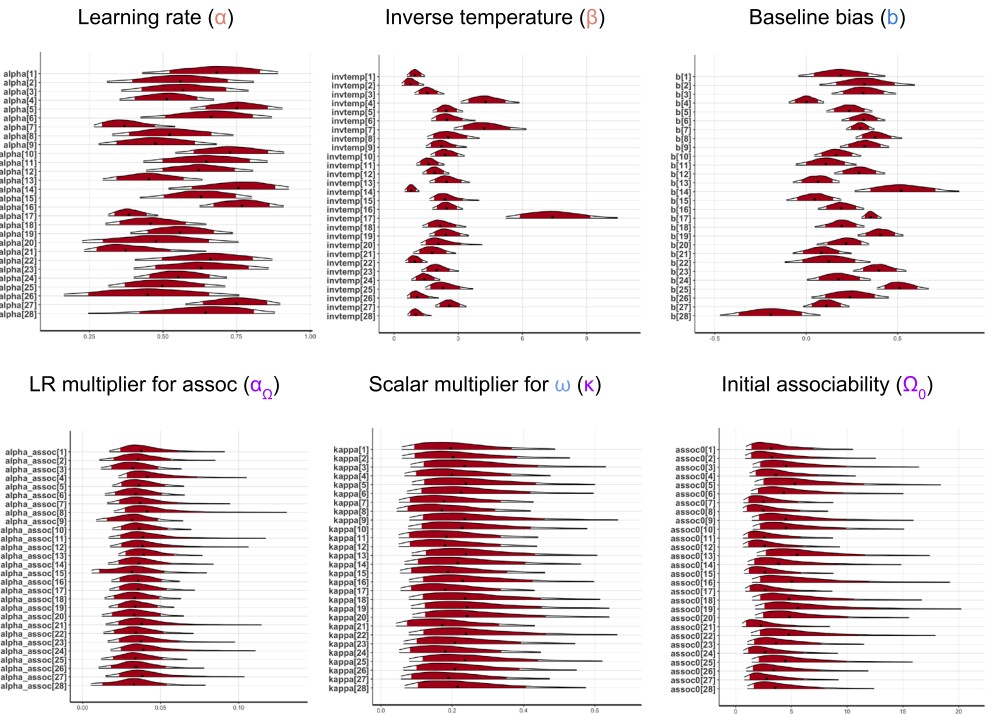

**Appendix 1—figure 8.** Subject-level parameter distributions from RL models. Subject-level parameter distributions from (**A**) the reinforcement learning (RL) model (M3) with fixed $\omega$ and (**B**) the RL model (M4) with flexible $\omega$. Shaded red regions denote 95% confidence intervals.

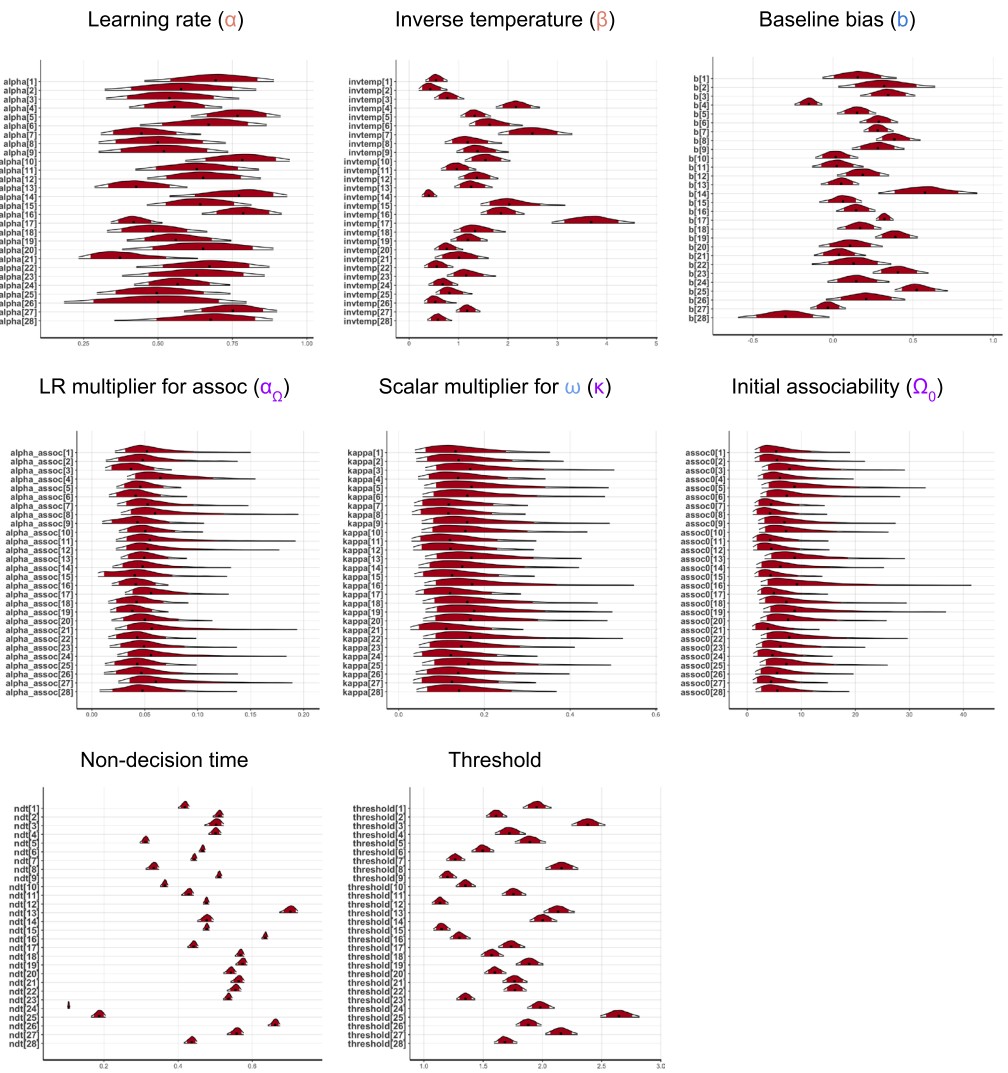

**Appendix 1—figure 9.** Subject-level parameter distributions from RLDDM models. Subject-level parameter distributions from (**A**) the reinforcement learning diffusion decision-making (RLDDM) model (M3) with fixed $\omega$ and (**B**) the RLDDM model (M4) with flexible $\omega$. Shaded red regions denote 95% confidence intervals.

## RL and RLDDM model parameters and model comparison tables

Please refer to *Appendix 1—table 1* and *Appendix 1—table 2*.

**Appendix 1—table 1.** Model comparison results for reinforcement learning (RL) models.

| Model | Free parameters | LOOIC | WAIC |
|---|---|---|---|
| M1 | $\alpha, \beta$ | 8201.53 | 8182.15 |
| M2 | $\alpha, \beta, b$ | 7960.00 | 7926.73 |
| M3 | $\alpha, \beta, b, \omega$ | 7947.84 | 7918.20 |
| M4 | $\alpha, \beta, b, \alpha_\Omega, \kappa, \Omega_0$ | 7863.79 | 7830.18 |

**Appendix 1—table 2.** Model comparison results for reinforcement learning diffusion decision-making (RLDDM) model.

| Model | Free parameters | LOOIC | WAIC |
|-------|-----------------|-------|------|
| M1 | ndt, threshold,$\alpha,\beta$ | 12539.14 | 12495.44 |
| M2 | ndt, threshold,$\alpha,\beta,b$ | 12303.00 | 12247.63 |
| M3 | ndt, threshold,$\alpha,\beta,b,\omega$ | 12296.38 | 12247.22 |
| M4 | ndt, threshold,$\alpha,\beta,b,\alpha_{\Omega},\kappa,\Omega_0$ | 12205.52 | 12164.14 |

## Model predictions: adapting fear responses in a chronic pain grid world

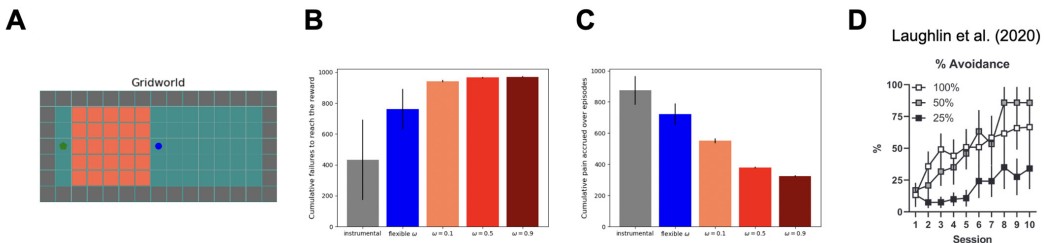

**Appendix 1—figure 10.** Model predictions: Adapting fear responses in a chronic pain gridworld. Pavlovian-instrumental interactions are invoked in a popular model of chronic pain, in which excessive Pavlovian fear of movement is self-punitive in a context in which active avoidance would reduce pain (*Crombez et al., 2012*; *Meulders et al., 2011*). (**A**) Grid world with a start at the centre (blue) and goal at the left end (green) operationalises this. We augment the action set to include an additional 'immobilise' action, to the action set, resulting in no state change and repeated rewards. An upper bound of 100 steps per episode is set; exceeding it leads to a painless death and episode restart. (**B**) Cumulative failures to reach the goal as a measure of efficiency. With a constant Pavlovian fear influence, the agent struggles to complete episodes, resembling effects seen in rodent models of anxiety (*Ligneul et al., 2022*). (**C**) Cumulative pain accrued as a measure of safety. In clinical terms, the agent remains stuck in a painful state, contrasting with an instrumental system that can seek and consume rewards despite pain. Flexible parameter $\omega$ ($\kappa = 3, \alpha_{\Omega} = 0.01$) allows the agent to overcome fear and complete episodes efficiently, demonstrating a safety-efficiency dilemma. The flexible $\omega$ policy outperforms fixed variants, emphasising the benefits of adapting fear responses for task completion. (**D**) Results from *Laughlin et al., 2020* show that 25% of the (anxious) rats fail the signalled active avoidance task due to freezing. GIFs for different configurations: pure instrumental agent, adaptively safe agent (flexible $\omega$), and maladaptively safe agent (constant $\omega$) can be found here.

# Neurobiology of Pavlovian contributions to bias avoidance behaviour

$a_p$ : Subset of actions with Pavlovian bias

$V_p(s)$: Pavlovian fear value

$\delta_p$: Pavlovian aversive prediction error

$Q(s,a)$: Instrumental values

$\delta$: Instrumental prediction errors

$a$ ~ softmax($\beta*\rho$): Action selection

$\Omega$ : Associability computation

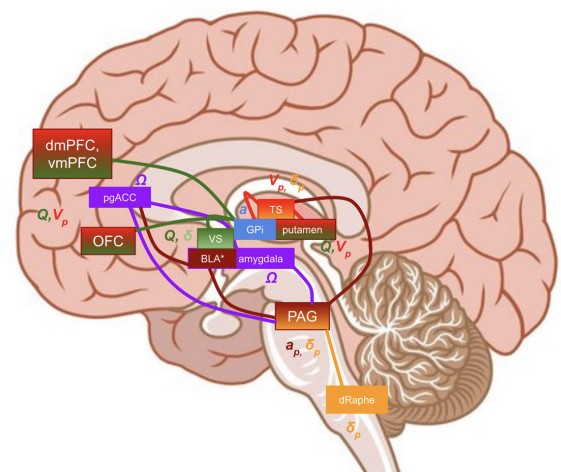

**Appendix 1—figure 11.** An overview of neurobiological substrates for the proposed Pavlovian avoidance learning (PAL) model based on relevant prior literature.

