## [Editor Report · eLife Assessment]

This **important** work describes results from a set of simulation and empirical studies of a set-up assessing exploratory behavior in a potentially rewarding environment that contains danger. The core idea is that an instrumental agent can be helped to be both effective and safe, thus avoiding excessive danger, during exploratory behavior, if the influence of an independent Pavlovian fear is flexibly gated based on uncertainty. This work is grounded in previous foundational work on Pavlovian control of instrumental choice, and significantly extends prior work showing that the impact of Pavlovian reward biases can be flexibly gated. The conclusion that safe but effective exploration can be achieved based on a flexibly weighted combination of a Pavlovian and an instrumental agent is **convincing**.

---

## [Referee Report · Reviewer #1 (Public review)]

Summary:

This paper provides a computational model of a synthetic task in which an agent needs to find a trajectory to a rewarding goal in a 2D-grid world, in which certain grid blocks incur a punishment. In a completely unrelated setup without explicit rewards, they then provide a model that explains data from an approach-avoidance experiment in which an agent needs to decide whether to approach, or withdraw from, a jellyfish, in order to avoid a pain stimulus, with no explicit rewards. Both models include components that are labelled as "Pavlovian"; hence the authors argue that their data show that the brain uses a "Pavlovian" fear system in complex navigational and approach-avoid decisions.

In the first setup, they simulate a model in which a "Pavlovian" component learns about punishment in each grid block, where as a Q-learner learns about the optimal path to the goal, using a scalar loss function for rewards and punishments. "Pavlovian" and Q-learning components are then weighed at each step to produce an action. Unsurprisingly, the authors find that including the "Pavlovian" component into the model reduces the cumulative punishment incurred, and this increases as the weight of the "Pavlovian" system increases. The paper does not explore to what extent increasing the punishment loss (while keeping reward loss constant) would lead to the same outcomes with a simpler model architecture.

In the second setup, an agent learns about punishments alone. So-called "Pavlovian biases" have previously been demonstrated in this task (i.e. an over avoidance when the correct decision is to approach). The authors explore several models to account for the Pavlovian biases.

Strengths:

Overall, the modelling exercises are interesting and relevant and incrementally expand the space of existing models.

Weaknesses:

For the first task, the simulation results are not compared to a simple Q-learning model. The second task is somewhat artificial, a problem compounded by the virtual reality setup. According to the cover story, participants get "stung by a jellyfish" on average 88 times during the experiment. In one condition, withdrawal from a jelly fish lead to a sting.

---

## [Referee Report · Reviewer #2 (Public review)]

Summary:

The authors tested the efficiency of a model combining Pavlovian fear valuation and instrumental valuation. This model is amenable to many behavioral decision and learning setups - some of which have been or will be designed to test differences in patients with mental disorders (e.g., anxiety disorder, OCD, etc.).

Strengths:

(1) Simplicity of the model which can at the same time model rather complex environments.

(2) Introduction of a flexible omega parameter.

(3) Direct application to a rather advanced VR task.

(4) The paper is extremely well written. It was a joy to read.

Weaknesses:

Almost none! In very few cases, the explanations could be a bit better.

Comments on revised version:

No further comments.

---

## [Referee Report · Reviewer #3 (Public review)]

Summary:

This paper aims to address the problem of exploring potentially rewarding environments that contain danger, based on the assumption that an independent Pavlovian fear learning system can help guide an agent during exploratory behaviour such that it avoids severe danger. This is important given that otherwise later gains seem to outweigh early threats, and agents may end up putting themselves in danger when it is advisable not to do so.

The authors develop a computational model of exploratory behaviour that accounts for both instrumental and Pavlovian influences, combining the two according to uncertainty in the rewards. The result is that Pavlovian avoidance has a greater influence when the agent is uncertain about rewards.

Strengths:

The study does a thorough job of testing this model using both simulations and data from human participants performing an avoidance task. Simulations demonstrate that the model can produce "safe" behaviour, where the agent may not necessarily achieve the highest possible reward but ensures that losses are limited. Interestingly, the model appears to describe human avoidance behaviour in a task that tests for Pavlovian avoidance influences better than a model that doesn't adapt the balance between Pavlovian and instrumental based on uncertainty. The methods are robust, and generally there is little to criticise about the study.

Weaknesses:

The methods are robust, and generally there is little to criticise about the study. The extent of the testing in human participants is fairly limited, but goes far enough to demonstrate that the model can account for human behaviour in an exemplar task. There are, however, some elements of the model that are unrealistic (for example, the fact that pre-training is required to select actions with a Pavlovian bias would require the agent to explore the environment initially and encounter a vast amount of danger in order to learn how to avoid the danger later), although this could simply reflect a lengthy evolutionary process.

---

## [Author Response]

The following is the authors’ response to the original reviews

**Reviewer #1 (Public review):**
Summary:This paper provides a computational model of a synthetic task in which an agent needs to find a trajectory to a rewarding goal in a 2D-grid world, in which certain grid blocks incur a punishment. In a completely unrelated setup without explicit rewards, they then provide a model that explains data from an approach-avoidance experiment in which an agent needs to decide whether to approach or withdraw from, a jellyfish, in order to avoid a pain stimulus, with no explicit rewards. Both models include components that are labelled as Pavlovian; hence the authors argue that their data show that the brain uses a Pavlovian fear system in complex navigational and approach-avoid decisions.

Thanks to the reviewer’s comments, we have now added the following text to our Discussion section (Lines 290-302):

“When it comes to our experiments, both the simulation and VR experiment models are related and derived from the same theoretical framework maintaining an algebraic mapping. They differ only in task-specific adaptations i.e. differ in action sets and differ in temporal difference learning rules - multi-step decisions in the grid world vs. Rescorla-Wagner rule for single-step decisions in the VR task. This is also true for Dayan et al. [2006] who bridge Pavlovian bias in a Go-No Go task (negative auto-maintenance pecking task) and a grid world task. A further minor difference between the simulation and VR experiment models is the use of a baseline bias in the human experiment's RL and the RLDDM model, where we also model reaction times with drift rates which is not a behaviour often simulated in the grid world simulations. As mentioned previously, we use the grid world tasks for didactic purposes, similar to Dayan et al. [2006] and common to test-beds for algorithms in reinforcement learning [Sutton et al., 1998]. The main focus of our work is on Pavlovian fear bias in safe exploration and learning, rather than on its role in complex navigational decisions. Future work can focus on capturing more sophisticated safe behaviours, such as escapes [Evans et al., 2019, Sporrer et. al., 2023] and model-based planning, which span different aspects of the threat-imminence continuum [Mobbs et al., 2020].”

In the first setup, they simulate a model in which a component they label as Pavlovian learns about punishment in each grid block, whereas a Q-learner learns about the optimal path to the goal, using a scalar loss function for rewards and punishments. Pavlovian and Q-learning components are then weighed at each step to produce an action. Unsurprisingly, the authors find that including the Pavlovian component in the model reduces the cumulative punishment incurred, and this increases as the weight of the Pavlovian system increases. The paper does not explore to what extent increasing the punishment loss (while keeping reward loss constant) would lead to the same outcomes with a simpler model architecture, so any claim that the Pavlovian component is required for such a result is not justified by the modelling.

Thanks to the reviewer’s comments, we have now added the following text to our Discussion section (Line 303-313):

“In our simulation experiments, we assume the coexistence of the Pavlovian fear system and the instrumental system to demonstrate the emergent safety-efficiency trade-off from their interaction. It is possible that similar behaviours could be modelled using an instrumental system alone, with higher punishment sensitivity, therefore we do not argue for the necessity for the Pavlovian fear system here. Instead, the Pavlovian fear system itself could be a potential biologically plausible implementation of punishment sensitivity. Unlike punishment sensitivity (scaling of the punishments), which has not been robustly mapped to neural substrates in fMRI studies; the neural substrates for the Pavlovian fear system are well known (e.g., the limbic loop and amygdala, further see Supplementary Fig. 16). Additionally, Pavlovian fear system provides a separate punishment memory that cannot be erased by greater rewards like [Elfwing and Seymour, 2017, Wang et al., 2018]. This fundamental point can be observed in our simple T-maze simulations, where the Pavlovian fear system encourages avoidance behaviour and the agent chooses the smaller reward instead of the greater reward.”

In the second setup, an agent learns about punishments alone. "Pavlovian biases" have previously been demonstrated in this task (i.e. an overavoidance when the correct decision is to approach). The authors explore several models (all of which are dissimilar to the ones used in the first setup) to account for the Pavlovian biases.

Thanks to the reviewer’s comments, we have now added a paragraph in our Discussion section (Line 290-302) explaining the similarity of our models and their integrated interpretation. We hope this addresses the reviewer’s concerns.

Strengths:Overall, the modelling exercises are interesting and relevant and incrementally expand the space of existing models.Weaknesses:I find the conclusions misleading, as they are not supported by the data.First, the similarity between the models used in the two setups appears to be more semantic than computational or biological. So it is unclear to me how the results can be integrated.

Thanks to the reviewer’s comments, we have now added a paragraph in our Discussion section (Line 290-302 onwards) explaining the similarity of our models and their integrated interpretation. We hope this addresses the reviewer’s concerns.

Secondly, the authors do not show "a computational advantage to maintaining a specific fear memory during exploratory decision-making" (as they claim in the abstract). Making such a claim would require showing an advantage in the first place. For the first setup, the simulation results will likely be replicated by a simple Q-learning model when scaling up the loss incurred for punishments, in which case the more complex model architecture would not confer an advantage. The second setup, in contrast, is so excessively artificial that even if a particular model conferred an advantage here, this is highly unlikely to translate into any real-world advantage for a biological agent. The experimental setup was developed to demonstrate the existence of Pavlovian biases, but it is not designed to conclusively investigate how they come about. In a nutshell, who in their right mind would touch a stinging jellyfish 88 times in a short period of time, as the subjects do on average in this task? Furthermore, in which real-life environment does withdrawal from a jellyfish lead to a sting, as in this task?Crucially, simplistic models such as the present ones can easily solve specifically designed lab tasks with low dimensionality but they will fail in higher-dimensional settings. Biological behaviour in the face of threat is utterly complex and goes far beyond simplistic fight-flight-freeze distinctions (Evans et al., 2019). It would take a leap of faith to assume that human decision-making can be broken down into oversimplified sub-tasks of this sort (and if that were the case, this would require a meta-controller arbitrating the systems for all the sub-tasks, and this meta-controller would then struggle with the dimensionality j).

Thanks to the reviewer’s comments, we have now mentioned this point in Lines 299-302.

On the face of it, the VR task provides higher "ecological validity" than previous screen-based tasks. However, in fact, it is only the visual stimulation that differs from a standard screen-based task, whereas the action space is exactly the same. As such, the benefit of VR does not become apparent, and its full potential is foregone.If the authors are convinced that their model can - then data from naturalistic approach-avoidance VR tasks is publicly available, e.g. (Sporrer et al., 2023), so this should be rather easy to prove or disprove. In summary, I am doubtful that the models have any relevance for real-life human decision-making.Finally, the authors seem to make much broader claims that their models can solve safety-efficiency dilemmas. However, a combination of a Pavlovian bias and an instrumental learner (study 1) via a fixed linear weighting does not seem to be "safe" in any strict sense. This will lead to the agent making decisions leading to death when the promised reward is large enough (outside perhaps a very specific region of the parameter space). Would it not be more helpful to prune the decision tree according to a fixed threshold (Huys et al., 2012)? So, in a way, the model is useful for avoiding cumulatively excessive pain but not instantaneous destruction. As such, it is not clear what real-life situation is modelled here.

We hope our additions to the Discussion section, from Line 290 to Line 313 address the reviewer’s concerns.

A final caveat regarding Study 1 is the use of a PH associability term as a surrogate for uncertainty. The authors argue that this term provides a good fit to fear-conditioned SCR but that is only true in comparison to simpler RW-type models. Literature using a broader model space suggests that a formal account of uncertainty could fit this conditioned response even better (Tzovara et al., 2018).

We have now added a line discussing this. (Line 356-358)

“Future work could also use a formal account of uncertainty which could fit the fear-conditioned skin-conductance response better than Pearce-Hall associability [Tzovara et al., 2018].”

**Reviewer #2 (Public review):**
Summary:The authors tested the efficiency of a model combining Pavlovian fear valuation and instrumental valuation. This model is amenable to many behavioral decision and learning setups - some of which have been or will be designed to test differences in patients with mental disorders (e.g., anxiety disorder, OCD, etc.).Strengths:(1) Simplicity of the model which can at the same time model rather complex environments.(2) Introduction of a flexible omega parameter.(3) Direct application to a rather advanced VR task.(4) The paper is extremely well written. It was a joy to read.Weaknesses:Almost none! In very few cases, the explanations could be a bit better.

Thank you, we have added further explanations in the discussion section. We have further improved the writing in abstract, introduction and Methods section taking into account recommendations from reviewer #2 and #3.

**Reviewer #2 (Recommendations for the authors):**
(1) Why is there no flexible omega in Figures 3B and 3C? Did I miss this?

Thank you. We have now added additional text to explain our motivation in Experiment 2, which only varies the fixed omega and omits the flexible omega (Lines 136-140).

“In this set of results, we wish to qualitatively tease apart the role of a Pavlovian bias in shaping and sculpting the instrumental value and also provide more insight into the resulting safety-efficiency trade-off. Having shown the benefits of a flexible ω in the previous section, here we only vary the fixed ω to illustrate the effect of a constant bias and are not concerned with the flexible bias in this experiment.”

We encourage the reader to consider this akin to an additional study that will explain how Pavlovian bias to withdraw can play a role in avoiding punishments similar to that of punishment sensitivity. This is particularly important as we do have neural correlates for Pavlovian biases but lack a clear neural correlation for punishment sensitivity so far, as mentioned in our new additions to the Discussion section (Lines 303-313).

(2) The introduction of the flexible omega and the PAL agent in the results is a bit sudden. Some more details are needed to understand this during the first read of this passage.

We thank reviewer #2 for bringing this to our notice. We have attempted to refine our passage by including sentences like -

“The standard (rational) reinforcement learning system is modelled as the instrumental learning system. The additional Pavlovian fear system biases the withdrawal actions to aid in safe exploration, in line with our hypothesis.”

“Both systems learn using a basic temporal difference updating rule (or in instances, its special case, the Rescorla-Wagner rule)”

“We implement the flexible ω using Pearce-Hall associability (see equation 15 in Methods). The Pearce-Hall associability maintains a running average of absolute temporal difference errors (δ) as per equation 14. This acts as a crude but easy-to-compute metric for outcome uncertainty which gates the influence of the Pavlovian fear system, in line with our hypothesis. This implies that higher the outcome uncertainty, as is the case in early exploration, the more cautious our agent will be, resulting in safer exploration”

(3) In my view, the possibility of modeling moving predators is extremely interesting. I would include Figure 8D and the corresponding explanation in the main text.

Response with revision: We thank the reviewer for finding our simulation on moving predators extremely interesting. Unfortunately, since our instrumental system is not model-based, and especially is not explicitly modelling the predator dynamics, our simulation might not be a very accurate representation of real moving predator environments. As pointed out by Reviewer #1, perhaps several other systems other than Pavlovian fear responses are necessary for safe behaviour in such environments and we hope to address these in future studies. Thanks again for taking an interest in our simulations.

(4) The VR experiment should be mentioned more clearly in the abstract and the introduction. It should be mentioned a bit more clearly why VR was helpful and why the authors did not use a simple bird's eye grid world task.I cannot assess the RLDDM and I did not check the code.

Thank you, we have now mentioned the VR experiment more clearly in the abstract and the introduction. We also now further mention that the VR experiment “builds upon previous Go-No Go studies studying Pavlovian-Instrumental transfer (Guitart-Masip et al, 2012; Cavanagh et al, 2013). The virtual-reality approach confers a greater ecological validity and the immersive nature may contribute better fear conditioning, making it easier to distinguish the aversive components.”

A bird’s eye grid world may not invoke a strong withdrawal response, as seen in these immersive approach-withdrawal tasks where we can clearly distinguish a Pavlovian fear-based withdrawal response. We did include immersive VR maze results in the supplementary materials, but future work is needed to isolate the different systems at play in such a complex behaviour.

**Reviewer #3 (Public review):**
Summary:This paper aims to address the problem of exploring potentially rewarding environments that contain the danger, based on the assumption that an independent Pavlovian fear learning system can help guide an agent during exploratory behaviour such that it avoids severe danger. This is important given that otherwise later gains seem to outweigh early threats, and agents may end up putting themselves in danger when it is advisable not to do so.The authors develop a computational model of exploratory behaviour that accounts for both instrumental and Pavlovian influences, combining the two according to uncertainty in the rewards. The result is that Pavlovian avoidance has a greater influence when the agent is uncertain about rewards.Strengths:The study does a thorough job of testing this model using both simulations and data from human participants performing an avoidance task. Simulations demonstrate that the model can produce "safe" behaviour, where the agent may not necessarily achieve the highest possible reward but ensures that losses are limited. Interestingly, the model appears to describe human avoidance behaviour in a task that tests for Pavlovian avoidance influences better than a model that doesn't adapt the balance between Pavlovian and instrumental based on uncertainty. The methods are robust, and generally, there is little to criticise about the study.Weaknesses:The extent of the testing in human participants is fairly limited but goes far enough to demonstrate that the model can account for human behaviour in an exemplar task. There are, however, some elements of the model that are unrealistic (for example, the fact that pre-training is required to select actions with a Pavlovian bias would require the agent to explore the environment initially and encounter a vast amount of danger in order to learn how to avoid the danger later). The description of the models is also a little difficult to parse.

Thank you, we have now attempted to clarify these points in the Discussion section by adding the following text (Lines 313-321):

“ We next discuss the plausibility of pre-training to select the hardwired actions In the human experiment, the withdrawal action is straightforwardly biased, as noted, while in the grid world, we assume a hardwired encoding of withdrawal actions for each state/grid. This innate encoding of withdrawal actions could be represented in the dPAG [Kim et al., 2013]. We implement this bias using pre-training, which we assume would be a product of evolution. Alternatively, this could be interpreted as deriving from an appropriate value initialization where the gradient over initialized values determines the action bias. Such aversive value initialization, driving avoidance of novel and threatening stimuli, has been observed in the tail of the striatum in mice, which is hypothesised to function as a Pavlovian fear/threat learning system [Menegas et al., 2018].”

**Reviewer #3 (Recommendations for the authors):**
I have relatively little to suggest, as in my view the paper is robust, thorough, and creative, and does enough to support the primary argument being made at the most fundamental level. My suggestions for improvement are as follows:(1) Some aspects of the model are potentially unrealistic (as described in the public review), and the paper may benefit from some discussion of these issues or attempts to make the model more realistic - i.e., to what extent is this plausible in explaining more complex avoidance behaviour? Primarily, the fact that pre-training is required to identify actions subject to Pavlovian bias seems unlikely to be effective in real-world situations - is there a better way to achieve this in cases where there isn't necessarily an instinctual Pavlovian response?

Thank you, we agree that the advantage of Pavlovian bias is restricted to the bias/instinctual Pavlovian response conferred by evolution. Future work is needed to model more complex avoidance behaviour such as escapes. We hope to have made this more clear with our edits to the Discussion (Lines 299-302) in our response to Reviewer #1’s comments, specifically:

“The main focus of our work is on Pavlovian fear bias in safe exploration and learning, rather than on its role in complex navigational decisions. Future work can focus on capturing more sophisticated safe behaviours, such as escapes [Evans et al., 2019, Sporrer et. al., 2023] and model-based planning which span different aspects of the threat-imminence continuum [Mobbs et al., 2020]”

(2) The description of the model in the method can be a little hard to follow and would benefit from further explanation of certain parameters. In general, it would be good to ensure that all terms mentioned in equations are described clearly in the text (for example, in Equation1 it isn't clear what k refers to).

Thank you, we have now added further information on all of the parameters in Equation 1 and overall improved the Methods section writing, for instance using time subscript for less confusion while introducing the parameters. We use the standard notation used in Sutton and Barto textbook. k refers to the timesteps into the future, and is now explained better in the Methods section.

(3) Another point of clarification in Equation 1 - does the policy account for the Pavlovian influence or is this purely instrumental?

Thank you, Equation 1 is purely instrumental. We have now specifically mentioned this. The Pavlovian influence follows later. They are combined into propensities for action as per equations 11-13.

(4) I was curious whether similar outcomes could be achieved by more complex instrumental models without the need for Pavlovian influences. For example, could different risk-sensitive decision rules (e.g., conditional value at risk) that rely only on the instrumental system afford safe behaviour without the need for an additional Pavlovian system?

Thank you for your comment. Yes, CVaR can achieve safe exploration/cautious behaviour in choices similar to Pavlovian avoidance learning. But we think both differ in the following ways:

(1) CVaR provides the correct solution to the wrong problem (objective that only maximises the lower tail of the distribution of outcomes)

(2) Pavlovian bias provides the wrong solution to the right problem (normative objective, but a Pavlovian bias which may be vestige of evolution)

Here we use the “wrong problem, wrong solution, wrong environment” categorisation terminology from Huys et al. 2015.

Huys, Q. J., Guitart-Masip, M., Dolan, R. J., & Dayan, P. (2015). Decision-theoretic psychiatry. Clinical Psychological Science, 3(3), 400-421.

Secondly, we find an effect of Pavlovian bias on reaction times - slowing down of approach responses and faster withdrawal responses. We do not think this can be best explained in a CVaR type model and is a direction for future work. We think such model-based methods are slower to compute, but Pavlovian withdrawal bias is quicker response.

We have now included this in brief in Lines 280-288.

(5) Figure 5 would benefit from a clearer caption as it is not necessarily clear from the current one that the left panels refer to choices and the right panels to reaction times.

Thank you, we have improved the caption for Fig. 5.

(6) It would be good to include some indication of the quality of the model fits for the human behavioural study (i.e., diagnostics such as R-hat) to ensure that differences in model fit between models are not due to convergence issues with different models. This would be especially helpful for the RLDDM models as these can be difficult to fit successfully.

Thank you, we observed that all Rhat values were strictly less than 1.05 (most parameters were less than 1.01 and generally close to 1), indicating that the models converged. We have now added this line to the results (Line 246-248). Thanks to the reviewer’s comments, we have now added the following text to our Discussion section (Lines 290-302): “When it comes to our experiments, both the simulation and VR experiment models are related and derived from the same theoretical framework maintaining an algebraic mapping. They differ only in task-specific adaptations i.e. differ in action sets and differ in temporal difference learning rules - multi-step decisions in the grid world vs. Rescorla-Wagner rule for single-step decisions in the VR task. This is also true for Dayan et al. [2006] who bridge Pavlovian bias in a Go-No Go task (negative auto-maintenance pecking task) and a grid world task. A further minor difference between the simulation and VR experiment models is the use of a baseline bias in the human experiment's RL and the RLDDM model, where we also model reaction times with drift rates which is not a behaviour often simulated in the grid world simulations. As mentioned previously, we use the grid world tasks for didactic purposes, similar to Dayan et al. [2006] and common to test-beds for algorithms in reinforcement learning [Sutton et al., 1998]. The main focus of our work is on Pavlovian fear bias in safe exploration and learning, rather than on its role in complex navigational decisions. Future work can focus on capturing more sophisticated safe behaviours, such as escapes [Evans et al., 2019, Sporrer et. al., 2023] and model-based planning, which span different aspects of the threat-imminence continuum [Mobbs et al., 2020].” In the first setup, they simulate a model in which a component they label as Pavlovian learns about punishment in each grid block, whereas a Q-learner learns about the optimal path to the goal, using a scalar loss function for rewards and punishments. Pavlovian and Q-learning components are then weighed at each step to produce an action. Unsurprisingly, the authors find that including the Pavlovian component in the model reduces the cumulative punishment incurred, and this increases as the weight of the Pavlovian system increases. The paper does not explore to what extent increasing the punishment loss (while keeping reward loss constant) would lead to the same outcomes with a simpler model architecture, so any claim that the Pavlovian component is required for such a result is not justified by the modelling.